# Erasing Undesirable Concepts in Diffusion Models with Adversarial Preservation

Anh Bui[1]    Long Vuong[1]    Khanh Doan[2]    Trung Le[1]    Paul Montague[3]
Tamas Abraham[3]    Dinh Phung[1]

[1]Monash University
[2]VinAI Research
[3]Defence Science and Technology Group, Australia

## Abstract

Diffusion models excel at generating visually striking content from text but can inadvertently produce undesirable or harmful content when trained on unfiltered internet data. A practical solution is to selectively removing target concepts from the model, but this may impact the remaining concepts. Prior approaches have tried to balance this by introducing a loss term to preserve neutral content or a regularization term to minimize changes in the model parameters, yet resolving this trade-off remains challenging. In this work, we propose to identify and preserving concepts most affected by parameter changes, termed as *adversarial concepts*. This approach ensures stable erasure with minimal impact on the other concepts. We demonstrate the effectiveness of our method using the Stable Diffusion model, showing that it outperforms state-of-the-art erasure methods in eliminating unwanted content while maintaining the integrity of other unrelated elements. Our code is available at `https://github.com/tuananhbui89/Erasing-Adversarial-Preservation`.

## 1   Introduction

Recent advances in text-to-image diffusion models (Rombach et al., 2022; Ramesh et al., 2021, 2022) have captured significant attention thanks to their outstanding image quality and boundless creative potential. These models undergo training on extensive internet datasets, enabling them to capture a wide range of concepts, which inevitably include undesirable concepts such as racism, sexism, and violence. Hence, these models can be exploited by users to generate harmful content, contributing to the proliferation of fake news, hate speech, and disinformation (Rando et al., 2022; Qu et al., 2023; Westerlund, 2019). Removing these undesirable contents from the model's output is thus a critical step in ensuring the safety and usefulness of these models.

Addressing this challenge, several methods have been proposed to erase undesirable concepts from pretrained text-to-image models, such as TIME (Orgad et al., 2023), UCE (Zhang et al., 2023), Concept Ablation (Kumari et al., 2023), and ESD (Gandikota et al., 2023). Despite differing approaches, these methods reach a common finding: removing even one concept can significantly reduce the model's ability to generate other concepts. This is because large-scale generative models $\mathcal{G} : \mathcal{C} \to \mathcal{X}$, such as Stable Diffusion (StabilityAI, 2022), are trained on billions of image-text pairs $(x, c)$, where $x$ is an image and $c$ is its caption, implicitly containing a set of concepts. The concept space is thus vast and intricately entangled within the model's parameters, meaning no specific part of the model's weights is solely responsible for a single concept. Consequently, the removal of one concept alters the entire model's parameters, causing a decline in overall performance. To address this degradation, existing methods typically select a neutral concept, such as "a photo" or an empty string, as an anchor to preserve while erasing the target concept, expecting that maintaining the neutral concept should help retain other concepts as well Orgad et al. (2023); Gandikota et al. (2024).

38th Conference on Neural Information Processing Systems (NeurIPS 2024).

While choosing a neutral concept is reasonable, we argue that it is not the optimal choice and may not guarantee the preservation of the model performance. In this paper, we propose to shift the attention towards the *adversarial concepts*, those most affected by changes in model parameters. This approach ensures that erasing unwanted content is stable and minimally impacts other concepts. To summarize, our key contributions are two-fold:

- We empirically investigate the impact of unlearning the target concept on the generation of other concepts. Our findings show that erasing different target concepts affects the remaining ones in various ways. This raises the question of whether preserving a neutral concept is sufficient to maintain the model's capability. We discover that the neutral concept lies in the middle of the sensitivity spectrum, whereas related concepts such as "person" and "women" are more sensitive to the target concept "nudity" than many neutral concepts. Additionally, we demonstrate that selecting the appropriate concepts to preserve significantly improves quality retention.

- We propose a novel method to identify the most sensitive concepts corresponding to the concept targeted to be erased, and then preserve these sensitive concepts explicitly to maintain the model's capability. We then conduct extensive experiments that demonstrate that the proposed method consistently outperforms other approaches in various settings.

## 2  Background of Text-to-Image Diffusion Models

**Denoising Diffusion Models:**   Generative modeling is a fundamental task in machine learning that aims to approximate the true data distribution $p_{\text{data}}$ from a dataset $\mathcal{D} = \{\mathbf{x}_i\}_{i=1}^N$. Diffusion models, a recent class of generative models, have shown impressive results in generating high-resolution images (Ho et al., 2020; Rombach et al., 2022; Ramesh et al., 2021, 2022). In a nutshell, training a diffusion model involves two processes: a forward diffusion process where noise is gradually added to the input image, and a reverse denoising diffusion process where the model tries to predict a noise $\epsilon_t$ which is added in the forward process. More specifically, given a chain of $T$ diffusion steps $x_0, x_1, ..., x_T$, the denoising process can be formulated as follows: $p_\theta(x_{T:0}) = p(x_T) \prod_{t=T}^1 p_\theta(x_{t-1} \mid x_t)$.

The model is trained by minimizing the difference between the true noise $\epsilon$ and $\epsilon_\theta(x_t, t)$, the predicted noise at step $t$ by the denoising model $\theta$ as follows:

$$\mathcal{L} = \mathbb{E}_{x_0 \sim p_{\text{data}}, t, \epsilon \sim \mathcal{N}(0, \mathbf{I})} \|\epsilon - \epsilon_\theta(x_t, t)\|_2^2 \tag{1}$$

**Latent Diffusion Models:**   With an intuition that semantic information that controls the main concept of an image can be represented in a low-dimensional space, (Rombach et al., 2022) proposed a diffusion process operating on the latent space to learn the distribution of the semantic information which can be formulated as $p_\theta(z_{T:0}) = p(z_T) \prod_{t=T}^1 p_\theta(z_{t-1} \mid z_t)$, where $z_0 \sim \varepsilon(x_0)$ is the latent vector obtained by a pre-trained encoder $\varepsilon$.

The objective function of the latent diffusion model as follows:

$$\mathcal{L} = \mathbb{E}_{z_0 \sim \varepsilon(x), x \sim p_{\text{data}}, t, \epsilon \sim \mathcal{N}(0, \mathbf{I})} \|\epsilon - \epsilon_\theta(z_t, t)\|_2^2 \tag{2}$$

## 3  Problem Statement

The task of erasing concepts from a text-to-image diffusion model often appears without additional data or labels, forcing us to rely on the model's own knowledge. Therefore, we here consider fine-tuning a pre-trained model rather than training a model from scratch. Let $\epsilon_\theta(z_t, c, t)$ denote the output of the pre-trained *foundation* U-Net model parameterized by $\theta$ at step $t$ given an input description $c \in \mathcal{C}$ and the latent vector from the previous step $z_t$ where $\mathcal{C}$ is set of all possible input descriptions, commonly referred to as the textual prompt in text-to-image generative models.

Given a set of textual descriptions $\mathbf{E} \subset \mathcal{C}$ and the target model $\epsilon_\theta$, our objective is to learn a *sanitized* model $\epsilon_{\theta'}(z_t, c, t)$ that cannot generate images from any textual description $c \in \mathbf{E}$ while preserving the quality of images generated by the remaining concepts $\mathcal{R} = \mathcal{C} \setminus \mathbf{E}$. We also use $c_n$ to denote a neutral or null concept, i.e., "a photo" or " ".

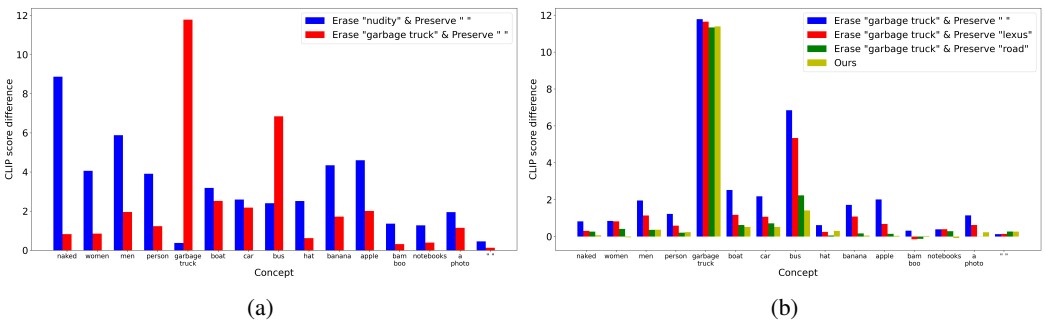

(a)                                                                          (b)

Figure 1: Analysis of the impact of erasing the target concept on the model's capability. The impact is measured by the difference of CLIP score $\delta(c)$ between the original model and the corresponding sanitized model. 1a: Impact of erasing "nudity" or "garbage truck" to other concepts. 1b: Comparing the impact of erasing the same "garbage truck" to other concepts with different preserving strategies, including preserving a fixed concept such as " ", "lexus", or "road", and adaptively preserving the most sensitive concept found by our method.

## 3.1 Naive Erasure

A naive approach that has been widely used in previous works Gandikota et al. (2023); Orgad et al. (2023); Gandikota et al. (2024) is to optimize the following objective function:

$$\min_{\theta'} \mathbb{E}_{c_e \in \mathbf{E}} \left[ \left\| \epsilon_{\theta'}(c_e) - \epsilon_\theta(c_n) \right\|_2^2 \right] \tag{3}$$

Fundamentally, these methods aim to force the model output, associated with the to-be-erased concepts, to approximate the model output associated with a neutral or null input $c_n$ (e.g., "a photo" or " "). Ideally, when erasing a concept, we would like to preserve *all* the remaining ones. This would corresponding to optimizing the above objective for all possible concepts in $\mathcal{C} \setminus \mathbf{E}$, which is excessively expensive. Hence, using a neutral concept as proxy first seems as a convenient strategy.

While this naive approach is effective in erasing the specific concept, it however has a negative impact on the model's capacity to preserve other concepts related to the to-be-erased concepts. For example, easing the concept "nudity" affects the quality of images of "woman" or "person". To mitigate this issue, prior works have proposed to use either an additional loss term to retain the null concept Gandikota et al. (2023) or a regularization term to prevent excessive change in the model parameters Orgad et al. (2023). However, these regularization attempts clearly have not addressed the core trade-off between erasing a concept and preserving the others.

## 3.2 Impact of Concept Removal on the Model Performance

We here approach the problem more carefully via a study on the impact of erasing a specific concept on model performance on the remaining ones. More importantly, we are concerned with the *most sensitive concepts* to erasure. For example, when removing the concept of "nudity", we are curious to know which concepts change the most in the model's output, so that we can preserve these concepts specifically to ensure the model's capability is maintained, at least with respect to these concepts.

For some concepts, we can make an intuitive guess. For example, the concept of "nudity" is closely related to the concepts of "women" and "men", which are likely to be affected by the removal of the concept of "nudity". However, for most concepts, it is not easy to determine which ones are most sensitive to the target concept. Therefore, in prior works, selecting a neutral one like a 'photo' or " " regardless of the target concepts is clearly not a sound solution. We next provide empirical evidence to support this argument.

**Measuring Generation Capability with CLIP Alignment Score.** Given a target concept $c_e$, e.g., "nudity" or "garbage truck", from that we obtain the original model $\epsilon_\theta$ and the sanitized model $\epsilon_{\theta'_{c_e}}$ by removing the target concept $c_e$. We have a set of concepts $\mathcal{C} = \{c_1, c_2, \ldots, c_{|\mathcal{C}|}\}$, where $|\mathcal{C}|$ is the number of concepts. Our goal is to measure the impact of unlearning $c_e$ on the generation of other concepts $c$ in $\mathcal{C}$.

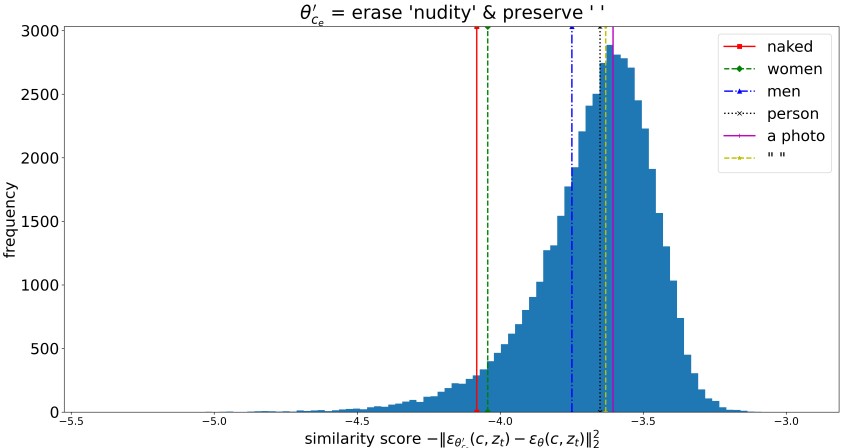

Figure 2: Sensitivity spectrum of concepts to the target concept "nudity". The histogram shows the distribution of the similarity score between outputs of the original model $\theta$ and the corresponding sanitized model $\theta'_{c_e}$ for each concept $c$ from the CLIP tokenizer vocabulary.

To achieve this, we generate a large number of samples from both models, i.e., $\{G(\theta, c, z_T^i)\}_{i=1}^k, \{G(\theta'_{c_e}, c, z_T^i)\}_{i=1}^k$ for $k = 200$ samples for each concept $c \in \mathcal{C}$. We then calculate the CLIP alignment score $S_{\theta,i,c} = S(G(\theta, c, z_T^i), c)$ between the generated samples and the textual description of the concepts $c$ (CLIP model 'openai/clip-vit-base-patch14'). A higher CLIP alignment score indicates that the generated samples are more similar to the concept $c$, and vice versa. Thus, we can use the CLIP alignment score as a metric to evaluate the capability of the model to generate the concept $c$, and the change of this score between the two models, $\delta_{c_e}(c) = \frac{1}{k} \sum_{i=1}^k \left( S_{\theta,i,c} - S_{\theta'_{c_e},i,c} \right)$ indicates the impact of unlearning $c_e$ on generating the concept $c$. The discussion on the metric is provided in Appendix B.3.

**The Removal of Different Target Concepts Leads to Different Side-Effects.** Figure 1a shows the impact of the removal of two distinct concepts, "nudity" and "garbage truck", on other concepts, measured by the difference of the CLIP score, $\delta_{\text{'nudity'}}(c), \delta_{\text{'garbage truck'}}(c)$. A larger $\delta(c)$ indicates a greater negative impact on the model's ability to generate concept $c$.

It can be seen that removing the "nudity" concept significantly affects highly related concepts such as "naked", "men", "women", and "person", while having minimal impact on unrelated concepts such as "garbage truck", 'bamboo' or neutral concepts such as "a photo" or the null " " concept. Similarly, removing the "garbage truck" concept significantly reduces the model's capability on concepts like "boat", "car", "bus", while also having little impact on other unrelated concepts such as "naked", "women" or neutral concepts.

These results suggest that removing different target concepts leads to varying impacts on other concepts. This indicates the need for an adaptive method to identify the most sensitive concepts relative to a particular target concept, rather than relying on random or fixed concepts for preservation. Moreover, in both cases, neutral concepts like "a photo" or the null concept show resilience and independence from changes in the model's parameters, suggesting that they do not adequately represent the model's capability to be preserved.

**Neutral Concepts lie in the Middle of the Sensitivity Spectrum.** Figure 2 shows the distribution of similarity scores between the outputs of the original model $\theta$ and the sanitized model $\theta'_{c_e}$ for each concept $c$ from the CLIP tokenizer vocabulary. The histogram reveals that the similarity scores span a wide range, indicating that the impact of unlearning the target concept on generating other concepts varies significantly. The lower the similarity score, the more different the outputs of the two models are, and the more sensitive the concept is to the target concept. Notably, the more related concepts like "women" or "men" are more sensitive to the removal of "nudity" than many neutral concepts that lie in the middle of the sensitivity spectrum.

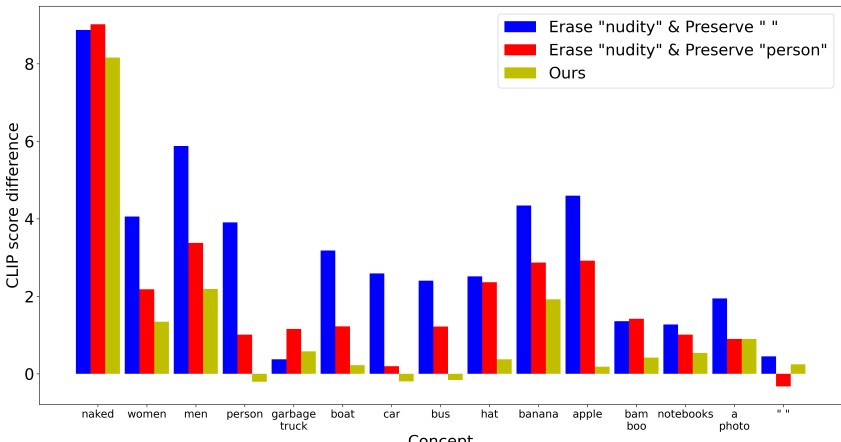

Figure 3: Comparing the impact of erasing the same "nudity" to other concepts with different preserving strategies.

**What Concept should be Kept to Maintain Model Performance.** Figure 1b presents the results of an experiment similar to the previous one, with one key difference: we utilize the prior knowledge gained from the previous experiment. Specifically, when erasing the "garbage truck", we apply different preservation strategies, including preserving a fixed concept such as " ", "lexus", or "road", and adaptively preserving the most sensitive concept found by our method.

The results show that with simple preservation strategies such as preserving a fixed but related concept like "road", the model's capability on other concepts is better maintained compared to preserving a neutral concept. However, the results of adaptively preserving the most sensitive concept show the best performance, with the least side effects on other concepts. Similarly, the results of erasing the "nudity" concept as shown in Figure 3 show that preserving related concepts like "person" helps retain the model's capability on other concepts much better than preserving a neutral concept. These findings confirm the importance of selecting sensitive concepts to preserve in order to better maintain the model's overall capability.

## 4 Proposed Method: Adversarial Concept Preservation

In this work, we aim to minimize the side effects of erasing undesirable concepts in diffusion models through adversarial preservation. Motivated by the observations in the previous section, our approach involves identifying the most sensitive concepts related to a specific target concept. For example, when removing the concept of nudity, we identify which concepts are most affected in the model's output so that we can specifically preserve these concepts to ensure the model's capability is maintained.

In each iteration, before updating the model parameters, we first identify the concept $c_a$ that is most sensitive to changes in the model parameters as we work to remove the target concepts.

$$\min_{\theta'} \max_{c_a \in \mathcal{R}} \mathbb{E}_{c_e \in \mathbf{E}} \left[ \underbrace{\|\epsilon_{\theta'}(c_e) - \epsilon_\theta(c_n)\|_2^2}_{L_1} + \lambda \underbrace{\|\epsilon_{\theta'}(c_a) - \epsilon_\theta(c_a)\|_2^2}_{L_2} \right] \qquad (4)$$

where $\lambda > 0$ is a parameter and $\mathcal{R} = \mathcal{C} \setminus \mathbf{E}$ denotes the remaining concepts.

Objective loss $L_1$ is the same as in the naive approach, aiming to erase the target concept $c_e$ by forcing its output to match that of a neutral concept. Our main contribution lies in the introduction of the adversarial preservation loss $L_2$, which aims to identify the most sensitive concept $c_a$ that is most affected by changes in the model parameters when removing the target concepts.

Since the concepts exist in a discrete space, the straightforward approach would involve revisiting all concepts in $\mathcal{R}$, resulting in significant computational complexity. Another naive approach is to consider the concepts as lying in a continuous space and use the Projected Gradient Descent (PGD)

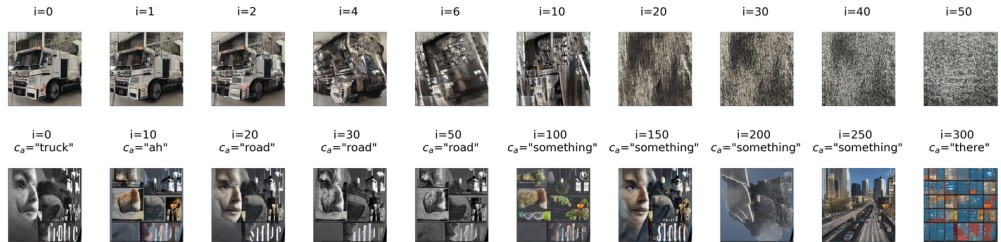

Figure 4: Images generated from the most sensitive concepts found by our method over the fine-tuning process. Top: Continous search with PGD. Bottom: Discrete search with Gumbel-Softmax. $c_a$ represents for the keyword.

method, similar to Madry et al. (2017), to search within the local region of the continuous space of the concepts. More specifically, we initialize the adversarial prompt with the text embedding of the to-be-erased concept, e.g., $c_{a,0} = c_e = \tau("\text{Garbage Truck}")$, and then update the adversarial concept with gradient $\nabla_{c_a} L_2$. Interestingly, while this approach provides an efficient computational method, we find that the adversarial concept quickly collapses from the initial concept to a background concept with the color information of the object as shown in the first row of Figure 4.

To combine the benefits of both approaches—making the process continuous and differentiable for efficient training while achieving meaningful concepts that are related to the target concept (second row of Figure 4)— we first define a distribution over the discrete concept embedding vector space as $\mathbb{P}_{\mathcal{R},\pi} = \sum_{i=1}^{|\mathcal{R}|} \pi_i \delta_{e_i}$ with the Dirac delta function $\delta$ and the weights $\pi \in \Delta_{\mathcal{R}} = \{\pi' \geq \mathbf{0} : \|\pi'\|_1 = 1\}$. Instead of directly searching for the most sensitive concept $c_a$ in the discrete concept embedding vector space $\mathcal{R}$, we switch to searching for the embedding distribution $\pi$ on the simplex $\Delta_{\mathcal{R}}$ and subsequently transform it back into a discrete space using the temperature-dependent GumbelSoftmax trick (Jang et al., 2016; Maddison et al., 2016) as follows:

$$\min_{\theta'} \max_{\pi \in \Delta_{\mathcal{R}}} \mathbb{E}_{c_e \in \mathbf{E}} \left[ \underbrace{\|\epsilon_{\theta'}(c_e) - \epsilon_\theta(c_n)\|_2^2}_{L_1} + \lambda \underbrace{\|\epsilon_{\theta'}(\mathbf{G}(\pi) \odot \mathcal{R}) - \epsilon_\theta(\mathbf{G}(\pi) \odot \mathcal{R})\|_2^2}_{L_2} \right] \tag{5}$$

where $\lambda > 0$ is a parameter, $\mathbf{G}$ is Gumbel-Softmax operator and $\odot$ is element wise multiplication operator. The pseudo-algorithm involves a two-step optimization process, outlined in Algorithm 1: *Finding Adversarial Concept* and Algorithm 2: *Adversarial Erasure Training*.

---

**Algorithm 1** Find Adversarial Concept

---

**Input:** $\theta, \mathcal{R}$. Searching hyperparameters: $\eta, N_{\text{iter}}$. Current state $\theta'_k$
**Output:** Adversarial concept $c_a$
**for** $i = 1$ to $N_{\text{iter}}$ **do**
    $\pi \leftarrow \pi + \eta \nabla_\pi \left[ \|\epsilon_{\theta'}(\mathbf{G}(\pi) \odot \mathcal{R}) - \epsilon_\theta(\mathbf{G}(\pi) \odot \mathcal{R})\|_2^2 \right]$       ▷ Maximize $L_2$
**end for**
$c_a = \mathbf{G}(\pi^*) \odot \mathcal{R}$

---

## 5 Experiments

In this section, we present a series of experiments to evaluate the effectiveness of our method in erasing various types of concepts from the foundation model. Our experiments use Stable Diffusion (SD) version 1.4 as the foundation model. We maintain consistent settings across all methods: fine-tuning the model for 1000 steps with a batch size of 1, using the Adam optimizer with a learning rate of $\alpha = 10^{-5}$. We benchmark our method against four baseline approaches: the original pre-trained SD model, ESD (Gandikota et al., 2023), UCE (Gandikota et al., 2024), and Concept Ablation (CA) (Kumari et al., 2023).

**Algorithm 2** Adversarial Erasure Training

---

**Input:** $\theta, \mathcal{R}, \mathbf{E}, \lambda$. Searching hyperparameters: $\eta, N_{\text{iter}}$.
**Output:** $\theta'$
$k \leftarrow 0, \theta'_k \leftarrow \theta$
**while** Not Converged **do**
    $c_e \sim \mathbf{E}$
    $c_a \leftarrow \text{FindAdversarialConcept}(\theta'_k, \theta, \mathcal{R}, \eta, N_{\text{iter}})$
    $\theta'_{k+1} \leftarrow \theta'_k - \alpha \nabla_{\theta'} [\|\epsilon_{\theta'}(c_e) - \epsilon_\theta(c_n)\|_2^2 + \lambda \|\epsilon_{\theta'}(c_a) - \epsilon_\theta(c_a)\|_2^2]$          ▷ Outer min
**end while**

---

We provide detailed implementation and further in-depth analysis in the appendix, including qualitative results (Section C), the choice of hyperparameters (Section B.2), and analysis on the search for the adversarial concepts (Sections B.4 and B.5).

## 5.1 Erasing Concepts Related to Physical Objects

In this experiment, we investigate the ability of our method to erase object-related concepts from the foundation model, for example, erasing entire object classes such as "Cassette Player" from the model. We choose Imagenette [1] which is a subset of the ImageNet dataset Deng et al. (2009) which comprises 10 easily recognizable classes, including "Cassette Player", "Chain Saw", "Church", "Gas Pump", "Tench", "Garbage Truck", "English Springer", "Golf Ball", "Parachute", and "French Horn".

Since the erasing performance when erasing a single class has been the main focus of previous work Gandikota et al. (2023), we choose a more challenging setting where we erase a set of 5 classes simultaneously. Specifically, we generate 500 images for each class and employ the pre-trained ResNet-50 He et al. (2016) to detect the presence of an object in the generated images. We use the two following metrics to evaluate the erasing performance: **Erasing Success Rate (ESR-k)**: The percentage of all the generated images with "to-be-erased" classes where the object is not detected in the top-k predictions. **Preserving Success Rate (PSR-k)**: The percentage of all the generated images with all other classes (i.e., "to-be-preserved") where the object is detected in the top-k predictions. This dual-metric evaluation provides a comprehensive assessment of our method's ability to effectively erase targeted object-related concepts while also preserving relevant elements.

**Quantitative Results.** We select four distinct sets of 5 classes from the Imagenette set for erasure and present the outcomes in Table 1. First, we note that the average PSR-1 and PSR-5 scores across the four settings of the original SD model stand at 78.0% and 97.6%, respectively. These scores indicate that 78.0% of the generated images contain the object-related concepts which are subsequently detected in the top-1 prediction, and when checking the concepts in any of the top-5 predictions, this number increases to 97.6%. This underscores the original SD model's ability to generate images with the anticipated object-related concepts.

In term of erasing performance, it can be observed that all baselines achieve very high ESR-1 and ESR-5 scores, with the lowest ESR-1 and ESR-5 scores being 95.5% and 88.9% respectively. This indicates the effectiveness of these methods to erase object-related concepts, as only a very small proportion of the generated images contain the object-related concepts under subsequent detection. Notably, the UCE method can achieve 100% ESR-1 and ESR-5, which is the highest among the baselines. Our method achieves 98.6% ESR-1 and 96.1% ESR-5, which is much higher than the two baselines ESD and CA, and only slightly lower than the UCE method, which is designed specifically for erasing object-related concepts.

However, despite the high erasing performance, the baselines, especially UCE, suffer from a significant drop in preserving performance, with the lowest PSR-1 and PSR-5 scores being 23.4% and 49.5%, respectively. This suggests that the preservation task poses greater challenges than the erasing task, and the baselines are ineffective in retaining other concepts. In contrast, our method achieves 55.2% PSR-1 and 79.9% PSR-5, which is a significant improvement compared to the best baseline, CA, with 44.2% PSR-1 and 66.5% PSR-5. This result underscores the effectiveness of our method in simultaneously erasing object-related concepts while preserving other unrelated concepts.

---

[1] https://github.com/fastai/imagenette

Table 1: Erasing object-related concepts.

| Method | ESR-1↑ | ESR-5↑ | PSR-1↑ | PSR-5↑ |
|--------|--------|--------|--------|--------|
| SD | $22.0 \pm 11.6$ | $2.4 \pm 1.4$ | $78.0 \pm 11.6$ | $97.6 \pm 1.4$ |
| ESD | $95.5 \pm 0.8$ | $88.9 \pm 1.0$ | $41.2 \pm 12.9$ | $56.1 \pm 12.4$ |
| UCE | $100 \pm 0.0$ | $100 \pm 0.0$ | $23.4 \pm 3.6$ | $49.5 \pm 8.0$ |
| CA | $98.4 \pm 0.3$ | $96.8 \pm 6.1$ | $44.2 \pm 9.7$ | $66.5 \pm 6.1$ |
| Ours | $98.6 \pm 1.1$ | $96.1 \pm 2.7$ | $55.2 \pm 10.0$ | $79.9 \pm 2.8$ |

## 5.2 Mitigating Unethical Content

One of the serious concerns associated with the deployment of text-to-image generative models to the public domain is their potential to generate Not-Safe-For-Work (NSFW) content. This ethical challenge has become a primary focus in recent works Schramowski et al. (2023); Gandikota et al. (2023, 2024), aiming to sanitize such capability of the model before public release.

In contrast to object-related concepts, such as "Cassette Player" or "English Springer", which can be explicitly described with limited textual descriptions, i.e., there are only a few textual ways to describe the visual concepts, unethical concepts like nudity are indirectly expressible in textual descriptions. The multiple ways a single visual concept can be described make erasing such concepts challenging, especially when relying solely on a keyword to indicate the concept to be erased. As empirically shown in Gandikota et al. (2023), the erasing performance on these concepts is highly dependent on the subset of parameters that are finetuned. Specifically, fine-tuning the non-cross-attention modules has shown to be more effective than fine-tuning the cross-attention modules. Therefore, in this experiment, we follow the same configuration as in Gandikota et al. (2023), focusing exclusively on fine-tuning the non-cross-attention modules.

**Quantitative Results.** To generate NSFW images, we employ I2P prompts Schramowski et al. (2023) and generate a dataset comprising 4703 images with attributes encompassing sexual, violent, and racist content. We then utilize the detector Praneet (2019) which can accurately detect several types of exposed body parts to recognize the presence of the nudity concept in the generated images. The detector Praneet (2019) provides multi-label predictions with associated confidence scores, allowing us to adjust the threshold and control the trade-off between the number of detected body parts and the confidence of the detection, i.e., the higher the threshold, the fewer the number of detected body parts.

Figure 5a illustrates the ratio of images with any exposed body parts detected by the detector Praneet (2019) over the total 4703 generated images (denoted by **NER**) across thresholds ranging from 0.3 to 0.8. Notably, our method consistently outperforms the baselines under all thresholds, showcasing its effectiveness in erasing NSFW content. In particular, as per Table 2, with the threshold set at 0.3, the NER score for the original SD model stands at 16.7%, indicating that 16.7% of the generated images contain signs of nudity concept from the detector's perspective. The two baselines, ESD and UCE, achieve 5.32% and 6.87% NER with the same threshold, respectively, demonstrating their effectiveness in erasing nudity concepts. Our method achieves 3.64% NER, the lowest among the baselines, indicating the highest erasing performance. This result remains consistent across different thresholds, emphasizing the robustness of our method in erasing NSFW content. Additionally, to measure the preserving performance, we generate images with COCO 30K prompts and measure the FID score compared to COCO 30K validation images. Our method achieves the best FID score of 15.52, slightly lower than that of UCE, which is the best baseline at 15.98, indicating that our method can simultaneously erase a concept while preserving other concepts effectively.

Detailed statistics of different exposed body parts in the generated images are provided in Figure 5b. It can be seen that in the original SD model, among all the body parts, the female breast is the most detected body part in the generated images, accounting for more than 320 images out of the total 4703 images. Both baselines, ESD and UCE, as well as our method, achieve a significant reduction in the number of detected body parts, with our method achieving the lowest number among the baselines. Our method also achieves the lowest number of detected body parts for the most sensitive body parts, only surpassing the baseline for less sensitive body parts, such as feet.

Interestingly, our method seems to remove the sensitive body parts while keeping the less sensitive body parts untouched as shown in Figure 5b. To provide more insights into this phenomenon, we

Table 2: Evaluation on the nudity erasure setting.

|  | NER-0.3↓ | NER-0.5↓ | NER-0.7↓ | NER-0.8↓ | FID↓ |
|---|---|---|---|---|---|
| CA | 13.84 | 9.27 | 4.74 | 1.68 | 20.76 |
| UCE | 6.87 | 3.42 | 0.68 | 0.21 | 15.98 |
| ESD | 5.32 | 2.36 | 0.74 | 0.23 | 17.14 |
| Ours | 3.64 | 1.70 | 0.40 | 0.06 | 15.52 |

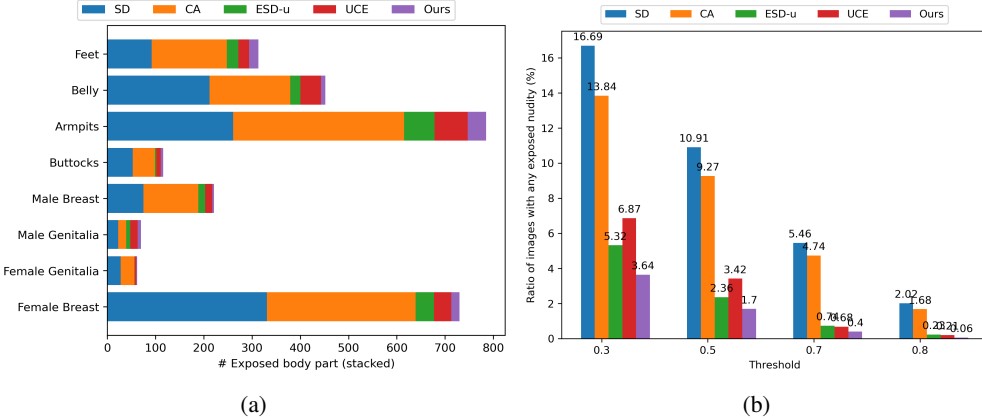

(a)             (b)

Figure 5: Comparison of the erasing performance on the I2P dataset. 5a: Number of exposed body parts counted in all generated images with threshold 0.5. 5b: Ratio of images with any exposed body parts detected by the detector Praneet (2019).

calculate the similarity scores between different concepts and body parts in the nudity erasure setting as Table 3.

Table 3: Similarity scores between different concepts and body parts in the nudity erasure setting.

| CLIP | Nudity | A photo | Person | Body |
|---|---|---|---|---|
| Feet | 0.612 | 0.547 | 0.566 | 0.643 |
| Belly | 0.601 | 0.514 | 0.517 | 0.748 |
| Armpits | 0.614 | 0.477 | 0.475 | 0.643 |
| Buttocks | 0.649 | 0.501 | 0.494 | 0.639 |
| Male Breast | 0.616 | 0.499 | 0.472 | 0.504 |
| Male Genitalia | 0.618 | 0.511 | 0.537 | 0.517 |
| Female Genitalia | 0.662 | 0.536 | 0.558 | 0.555 |
| Female Breast | 0.656 | 0.517 | 0.491 | 0.574 |

It can be seen that the "nudity" concept is highly correlated with the "Female Breast" concept, suggesting that when removing the "nudity" concept, the "Female Breast" concept is more likely to be affected than other body parts. On the other hand, the "Person" or "Body" concept is more strongly correlated with the "Feet" concept than with the "Female Breast" concept, indicating that preserving the "Person" concept might help maintain the model's performance on "Feet" rather than on "Female Breast." Furthermore, the gap between the "Feet" and "Female Breast" concepts with respect to "Person" or "Body" is larger than the gap with more generic concepts like "A photo." This suggests that preserving generic concepts might not have the same impact as preserving the most affected concepts. Our method naturally selects the most affected concepts to be preserved, which often includes concepts highly correlated with non-sensitive body parts. This explains the observed phenomenon in the experiment.

## 5.3 Erasing Artistic Concepts

In this experiment, we investigate the ability of our method to erase artistic style concepts from the foundation model. We choose several famous artists with easily recognizable styles who have been known to be mimicked by the text-to-image generative models, including "Kelly Mckernan", "Thomas Kinkade", "Tyler Edlin" and "Kilian Eng" as in Gandikota et al. (2023). We compare our

method with recent work including ESD Gandikota et al. (2023), UCE Gandikota et al. (2024), and CA Kumari et al. (2023) which have demonstrated effectiveness in similar settings.

For fine-tuning the model, we use only the names of the artists as inputs. For evaluation, we use a list of long textual prompts that are designed exclusively for each artist, combined with 5 seeds per prompt to generate 200 images for each artist across all methods. We measure the CLIP alignment score [2] between the visual features of the generated image and its corresponding textual embedding. Compared to the setting Gandikota et al. (2023) which utilized a list of generic prompts, our setting with longer specific prompts can leverage the CLIP score as a more meaningful measurement to evaluate the erasing and preserving performance. We also use LPIPS Zhang et al. (2018) to measure the distortion in generated images by the original SD model and editing methods, where a low LPIPS score indicates less distortion between two sets of images.

It can be seen from Table 4 that our method achieves the best erasing performance while maintaining a comparable preserving performance compare to the baselines. Specifically, our method attains the lowest CLIP score on the to-be-erased sets at 21.57, outperforming the second-best score of 23.56 achieved by ESD. Additionally, our method secures a 0.78 LPIPS score, the second-highest, following closely behind the CA method with 0.82. Concerning preservation performance, we observe that, while our method achieves a slightly higher LPIPS score than the UCE method, suggesting some alterations compared to the original images generated by the SD model, the CLIP score of our method remains comparable to these baselines. This implies that our generated images still align well with the input prompt.

Table 4: Erasing artistic style concepts.

|  | To Erase | | To Retain | |
|  | CLIP$\downarrow$ | LPIPS$\uparrow$ | CLIP$\uparrow$ | LPIPS$\downarrow$ |
|---|---|---|---|---|
| ESD | $23.56 \pm 4.73$ | $0.72 \pm 0.11$ | $29.63 \pm 3.57$ | $0.49 \pm 0.13$ |
| CA | $27.79 \pm 4.67$ | $0.82 \pm 0.07$ | $29.85 \pm 3.78$ | $0.76 \pm 0.07$ |
| UCE | $24.47 \pm 4.73$ | $0.74 \pm 0.10$ | $30.89 \pm 3.56$ | $0.40 \pm 0.13$ |
| Ours | $21.57 \pm 5.46$ | $0.78 \pm 0.10$ | $30.13 \pm 3.44$ | $0.47 \pm 0.14$ |

# 6 Conclusion

In this paper, we introduced a novel approach to concept erasure in text-to-image diffusion models by incorporating an adversarial learning mechanism. This mechanism identifies the most sensitive concepts affected by the removal of the target concept from the discrete space of concepts. By preserving these sensitive concepts, our method outperforms state-of-the-art erasure techniques in both erasing unwanted content and preserving unrelated concepts, as demonstrated through extensive experiments. Furthermore, our adversarial learning mechanism exhibits high flexibility, linking this task to the field of Adversarial Machine Learning, where adversarial examples have been extensively studied. This connection opens potential directions for future research, such as simultaneously searching for multiple sensitive concepts under certain divergence constraints, offering promising avenues for further exploration.

## Acknowledgements

This work was supported by the Australian Defence Science and Technology (DST) Group through the Next Generation Technology Fund (NGTF) scheme and the Department of Defence, Australia, via the Advanced Strategic Capabilities Accelerator (ASCA) program. Dinh Phung further acknowledged the support from the Australian Research Council (ARC) Discovery Project DP230101176. The authors would like to express their appreciation to the anonymous reviewers for their insightful feedback and valuable suggestions, which has significantly enhanced the quality of this work.

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

# Appendix

## Table of Contents

## A  Related Work

Given the growing concerns over the potential misuse of text-to-image models, several techniques have been developed to remove undesirable concepts from foundation models before deployment. The simplest approach is ***pre-processing***, which filters out objectionable content from the training data using pre-trained detectors. This method, as seen in Stable Diffusion v2.0 (StabilityAI, 2022) and Dall-E[3], excludes harmful data from the training set. However, it requires retraining the entire model, making it computationally expensive and impractical for adapting to evolving erasure requests.

Another basic approach is ***post-processing***, which aims to identify potentially inappropriate content in generated data and then either blur or black out the images before they are presented to users. This method involves a Not-Safe-For-Work (NSFW) detector, which can be deployed along with the generative model, as seen in closed-source models like Dall-E or Midjourney, or released as a separate module in open-source models like Stable Diffusion. However, this approach is not foolproof, as demonstrated in (Yang et al., 2024), where a technique similar to the Boundary Attack (Brendel et al., 2017) was used to uncover adversarial prompts that could bypass the filtering mechanism. In the case of open-source models, the NSFW detector can be easily disabled by modifying just a few lines of code in the source (SmithMano, 2022).

To date, the most successful strategy for sanitizing open-source models, such as Stable Diffusion, is ***model fine-tuning***, which involves sanitizing the generator (e.g., U-Net) in the diffusion model post-training on raw, unfiltered data and before public release. This approach, as partially demonstrated in Gandikota et al. (2023, 2024), underscores the importance of addressing potential biases and undesired content in models before their deployment. There are two main branches within model fine-tuning: attention-based and output-based, categorized by the primary components involved in the objective function.

**Attention-based** methods (Zhang et al., 2023; Orgad et al., 2023; Kumari et al., 2023; Gandikota et al., 2024; Lu et al., 2024) focus on modifying the attention mechanisms within models to remove undesirable concepts. In Latent Diffusion Models (LDMs), for instance, the textual conditions are embedded via a pre-trained CLIP model and injected into the cross-attention layers of the UNet model (Rombach et al., 2022; Ramesh et al., 2022). Therefore, removing an unwanted concept can be achieved by altering the attention mechanism between the textual condition and visual information flow. For example, in TIME (Orgad et al., 2023), the authors propose to minimize $\|W^{'} c_e - v_t^*\|_2^2$, where $W$ represents the original cross-attention weights, $W^{'}$ the fine-tuned weights, $c_e$ the embedding of the unwanted concept, and $v_t^*$ the target vector. By different settings of $v_t^*$, the method can either steer the unwanted concept toward a more acceptable one (i.e., $v_t^* = W\tau(\text{"a photo"})$) or edit biases

---

[3]https://openai.com/index/dall-e-2-pre-training-mitigations/

in the model (i.e., $v_t^* = W\tau(\text{"a female doctor"})$). This category has two main advantages including the closed-form solution as shown in (Orgad et al., 2023), and the fact that it operates solely on textual embeddings not the intermediate images, making it faster than optimization-based methods.

Follow-up works (Zhang et al., 2023; Gandikota et al., 2024; Lu et al., 2024) share this principle. Specifically, Forget-Me-Not (Zhang et al., 2023) introduces an attention resteering method that minimizes the L2 norm of the attention maps related to the unwanted concept. UCE (Gandikota et al., 2024) extends TIME by proposing a preservation term that allows the retention of certain concepts while erasing others. MACE (Lu et al., 2024) improves the generality and specificity of concept erasure by employing LoRA modules (Hu et al., 2021) for each individual concept, combining them with the closed-form solution from TIME (Orgad et al., 2023).

*Output-based* methods (Gandikota et al., 2023; Bui et al., 2024) focus on optimizing the output image by minimizing the difference between the predicted noise $\epsilon_{\theta'}(z_t, t, c_e)$ and the target noise $\epsilon_\theta(z_t, t, c_t)$. Unlike attention-based methods, this approach requires intermediate images $z_t$ sampled at various time steps $t$ during the diffusion process. While this method is computationally more expensive, it generally yields superior erasure results by directly optimizing the image, ensuring the removal of unwanted concepts (Gandikota et al., 2023).

A recent addition to the field, SPM (Lyu et al., 2024), introduces one-dimensional adapters that, when combined with pre-trained LDMs, prevent the generation of images containing unwanted concepts. SPM introduces a new diffusion process $\hat{\epsilon} = \epsilon(x_t, c_t \mid \theta, \mathcal{M}_{c_e})$, where $\mathcal{M}_{c_e}$ is an adapter model trained to remove the undesirable concept $c_e$. While these adapters can be shared and reused across different models, the original model $\theta$ remains unchanged, allowing malicious users to remove the adapter and generate harmful content. Thus, SPM is less robust and practical compared to the other approaches discussed.

***Concept mimicry*** is a recent research direction that aims to copy or mimic a specific concept from a set of reference images to generate new images containing the concept. The concept can be artistic styles or personal visual appearance, raising concerns about the potential misuse of the technique. Noteworthy methods include Textual Inversion (Gal et al., 2022) and Dreambooth (Ruiz et al., 2023), which have proven effective with just a few user-provided images. In contrast, ***Anti Concept Mimicry*** is employed to safeguard personal or artistic styles from being copied through concept mimicry. Achieved by introducing imperceptible adversarial noise to input images, this technique can deceive concept mimicry methods under specific conditions. Recent contributions such as Anti-Dreambooth (Van Le et al., 2023), SDS (Xue et al., 2023), and MetaCloak (Liu et al., 2024) have explored and demonstrated the effectiveness of this approach. EditGuard (Zhang et al., 2024a), on the other hand, aims to watermark images with imperceptible adversarial noise to localize tampered regions and claim copyright protection. This category can be viewed as a *protection method from the user's side*, which is orthogonal to the erasure problem discussed in this paper.

Developed concurrently with this paper, ***AdvUnlearn*** (Zhang et al., 2024b) incorporates adversarial training (Goodfellow et al., 2014; Madry et al., 2017; Bui et al., 2020, 2022) to improve the robustness of concept erasure. More specifically, the authors propose a similar bilevel min-min optimization problem, where the inner minimization problem seeks to find the adversarial prompt $c_a$ that minimizes the attack loss, i.e., the extent to which the unwanted concept is retained in the generated image. The adversarial prompt $c_a$ is found using adversarial prompt attack techniques (Zhang et al., 2025; Chin et al., 2023), such as the fast gradient sign method (FGSM) (Goodfellow et al., 2014).

While both *AdvUnlearn* and our approach share the adversarial training framework, they are fundamentally different. Firstly, our method is driven by the observation of how erasure impacts model performance and how different preservation strategies affect erasure efficacy. *AdvUnlearn*, on the other hand, is motivated by adversarial prompt attacks. Secondly, our method formulates the problem as a bilevel min-max optimization, where the inner maximization aims to find the adversarial concept $c_a$ that maximizes the preservation loss, while *AdvUnlearn*'s inner minimization seeks the adversarial prompt $c^*$ that minimizes the attack loss. Finally, our method employs the Gumbel-Softmax trick (Jang et al., 2016) to approximate the bilevel optimization, whereas *AdvUnlearn* uses FGSM to find the adversarial prompt.

# B  Further Experiments

## B.1  Experimental Settings

**General Settings.**  Our experiments use Stable Diffusion (SD) version 1.4 as the foundation model. We maintain consistent settings across all methods, fine-tuning the model for 1000 steps with a batch size of 1, using the Adam optimizer with a learning rate of $1e-5$. We benchmark our method against four baseline approaches: the original pre-trained SD model, ESD (Gandikota et al., 2023), UCE (Gandikota et al., 2024), and Concept Ablation (CA) (Kumari et al., 2023). Our models are trained on 1 NVIDIA A100 GPUs of 80GB. One training routine takes less than 6 hours for erasing "nudity" and less than 1 hour for other concepts.

**Settings for Our Method.**  A crucial aspect of our method is the concept space $\mathcal{R}$, where we search for the most sensitive concept $c_a$. In our experiments, we use two vocabularies: the CLIP token vocabulary, which includes 49,408 tokens, and the Oxford 3000 word list, comprising the 3000 most common English words[4]. While the CLIP token vocabulary is more comprehensive, it presents challenges due to the large number of nonsensical tokens. Therefore, for the experiments in Section 5, we use the Oxford 3000-word list to demonstrate the effectiveness of our method.

**Computational Limitations.**  To search for the adversarial concept $c_a$ effectively, we employ the Gumbel-Softmax trick (Jang et al., 2016) to sample from the categorical distribution in the concept space $\mathcal{R}$. This approach requires feeding the model with the embeddings of the entire concept space $\mathcal{R}$, which exponentially increases the computational cost as the size of the concept space grows. To mitigate this, we use a subset of the $K$ most similar concepts to the target concept $c_e$ to reduce computational costs. The similarity between concepts is calculated using cosine similarity between their embeddings.

To provide a better understanding of the concept space $\mathcal{R}$, we list the $K$ most similar concepts to the target concept $c_e$ in Table 5. We provide the study of the impact of the number of concepts $K$ and the number of search steps $N_{\text{iter}}$ on the erasing and preservation performance in Section B. It is worth to remind that, erasing "nudity" requires to fine-tune on all non-cross-attention modules which is more computationally expensive than erase other concepts that only requires fine-tuning on cross-attention modules. Therefore, in the default settings, we use $K = 50$ for erasing 'nudity' and $K = 100$ for other concepts. For searching hyperparameters, we use $N_{\text{iter}} = 2$, $\eta = 1 \times 10^{-3}$, and a trade-off $\lambda = 1$ as the default settings.

## B.2  Impact of Hyperparameters

In this section, we investigate the impact of hyperparameters on the performance of our method. Specifically, we analyze the effect of the number of closest concepts $K$ and the number of search steps $N_{\text{iter}}$ on the erasing and preservation performance. We conduct the experiments on the Imagenette dataset with the same settings as in Section 5. Table 6 shows the evaluation results of different hyperparameter settings. It can be seen that the erasing and preservation performance is more affected by the number of concepts $K$ than the number of search steps $N_{\text{iter}}$. Reducing the search space from $K = 100$ to $K = 20$ hugely decreases the erasing performance by around 2% in ESR-1 and 3% in ESR-5, as well as the preservation performance by around 2% in PSR-5. This observation aligns with the intuition that a larger search space provides more flexibility for the model to find the most sensitive concept $c_a$.

On the other hand, increasing the number of search steps from $N_{\text{iter}} = 2$ to $N_{\text{iter}} = 8$ does not increase the performance, but inversely hurts the preservation performance by around 10% in PSR-1. Therefore, in other experiments, we use $K = 100$ and $N_{\text{iter}} = 2$ as the default settings.

**Impact of the Concept Space**  To ensure the generality of the search space so that it can be applied to various tasks such as object-related concepts, NSFW content, and artistic styles, we used the Oxford 3000 most common words in English as the search space.

To evaluate the impact of the concept space, we conduct additional experiments with the search space as the CLIP token vocabulary, which includes 49,408 tokens. It is worth noting that the CLIP token

---

[4]https://www.oxfordlearnersdictionaries.com/wordlist/american_english/oxford3000/

Table 5: The list of the $K = 50$ most similar concepts to the target concept $c_e$.

| Target Concept | Similar Concepts |
| --- | --- |
| **Garbage truck** | truck, vehicle, waste, pollution, bus, terrible, container, vegetable, something, awful, refuse, delivery, destruction, that, transportation, another, traffic, engine, grocery, machine, comprehensive, divorce, dirty, interesting, organic, great, opinion, typical, well, yeah, really, stupid, controversy, painful, object, funny, garage, political, sick, neighborhood, sentence, deliver, interpretation, again, disaster, poverty, complaint, apparent, regarding, continued |
| **Cassette player** | music, speaker, tape, radio, telephone, instrument, musical, technology, electronic, phone, musician, communication, classic, battery, opposite, listen, topic, phrase, volume, sound, television, object, exchange, memory, item, record, motor, introduce, theme, communicate, cognitive, machine, context, rhythm, subject, comprehensive, contest, interpretation, camera, historian, love, player, equipment, regarding, definition, historical, hello, description, creative, chapter |
| **Parachute** | air, bag, tent, fabric, drop, above, day, swing, at, personal, helicopter, open, new, part, fairly, current, package, plane, under, and, second, fourth, another, then, first, far, favorite, from, opening, float, valley, modest, low, just, fun, third, patch, string, along, slightly, catch, flight, my, little, top, unusual, recent, launch, in, near |
| **Church** | church, religious, Catholic, religion, another, town, Christian, prayer, spiritual, priest, hospital, clinic, neighborhood, bank, museum, previous, again, newly, village, faith, where, various, last, rural, yesterday, holy, court, first, funeral, continued, recent, then, love, factory, today, sacred, cross, near, there, more, place, second, farm, school, from, something, past, porch, store, around |
| **French horn** | French, another, theme, cycle, instrument, rhythm, there, again, then, composition, first, musical, music, afternoon, compose, forth, third, bell, musician, circumstance, portion, borrow, comprehensive, continued, that, hour, around, second, sound, although, assessment, while, last, proposed, fourth, fun, along, telescope, slightly, this, just, yeah, previous, though, headline, hear, arrangement, definition, addition, brief |
| **Nudity** | naked, sex, hot, sexual, breast, modest, nut, and, dirty, skin, from, second, interested, physically, new, curious, also, third, just, enjoy, then, another, my, good, at, first, in, bad, current, day, kind, body, slightly, lovely, quite, recent, interesting, so, show, episode, near, full, primarily, unique, particularly, reveal, oh, ah, wide, today |
| **Kelly McKernan** | voice, between, put, emotional, blue, immediate, flesh, sweet, primarily, pale, newly, I, combination, currently, spending, fresh, consistent, provide, among, pair, international, teen, soul, income, him, kind, catch, feel, attractive, sister, pretty, fun, any, thin, and, good, inner, naturally, natural, recent, embrace, could, investigation, make, beneath, rough, post, attitude, lover, luck |
| **Ajin Demi Human** | angle, morning, prepare, human, eight, my, order, possible, hi, trace, democratic, unknown, should, political, remain, via, ill, identification, designer, new, story, heart, very, tension, nearly, just, perfect, medical, friendly, protein, hello, poor, killer, all, although, racial, thanks, religion, beginning, definition, juice, Arab, hot, senator, and, main, figure, prisoner, day, the |

Table 6: Evaluation of the impact of hyperparameters on the erasing and preservation performance.

| Method | ESR-1↑ | ESR-5↑ | PSR-1↑ | PSR-5↑ |
|---|---|---|---|---|
| $K = 100, N_{\text{iter}} = 2$ | 98.72 | 95.60 | 63.80 | 82.96 |
| $K = 100, N_{\text{iter}} = 4$ | 98.64 | 95.84 | 63.76 | 80.72 |
| $K = 100, N_{\text{iter}} = 8$ | 98.84 | 95.96 | 54.12 | 70.96 |
| $K = 20, N_{\text{iter}} = 2$ | 96.76 | 92.76 | 63.28 | 80.88 |
| $K = 50, N_{\text{iter}} = 2$ | 96.92 | 91.48 | 63.32 | 81.40 |
| $K = 100, N_{\text{iter}} = 2$ | 98.72 | 95.60 | 63.80 | 82.96 |

vocabulary is more comprehensive but presents challenges due to the large number of nonsensical tokens (e.g., "...", "."</w>" ). Therefore, we need to filter out these nonsensical tokens to ensure the quality of the search space. The results from object-related concepts are shown in the table below.

Table 7: Evaluation of the impact of the concept space on the erasing and preservation performance.

| Vocab | ESR-1↑ | ESR-5↑ | PSR-1↑ | PSR-5↑ |
|---|---|---|---|---|
| Oxford | 98.72 | 95.60 | 63.80 | 82.96 |
| CLIP | 97.88 | 94.80 | 69.24 | 87.20 |

The results in Table 7 show that the erasing performance is slightly lower when using the CLIP token vocabulary as the search space, but the preservation performance is much better with a gap of 5.4% in PSR-1 and 4.2% in PSR-5. This indicates that the quality of the search space is a crucial factor for the performance of our method, and different tasks might require customized search spaces to achieve better performance.

**Choosing the model's parameters for fine-tuning.** Firstly, it is a worth recall that the cross-attention mechanism, i.e., $\sigma(\frac{(QK^T)}{\sqrt{d}})V$, where $Q$, $K$, and $V$ are the query, key, and value matrices, respectively. In text-to-image diffusion models like SD, the key and value are derived from the textual embedding of the prompt, while the query comes from the previous denoising step. The cross-attention mechanism allows the model to focus on the relevant parts of the prompt to generate the image.

Therefore, when unlearning a concept, most of the time, the erasure process is done by loosening the attention between the query and the key that corresponds to the concept to be erased, i.e., by fine-tuning the cross-attention modules. This approach works well for object-related concepts or artistic styles, where the target concept can be explicitly described with limited textual descriptions.

However, as investigated in the ESD paper Section 4.1 (Gandikota et al., 2023), concepts like 'nudity' or NSFW content can be described in various ways, many of which do not contain explicit keywords like 'nudity.' This makes it inefficient to rely solely on keywords to indicate the concept to be erased. It is worth noting that the standard SD model has 12 transformer blocks, each of which contains one cross-attention module but also several non-cross-attention modules such as self-attention and feed-forward modules, not to mention other components like residual blocks. Therefore, fine-tuning the non-cross-attention modules will have a more global effect on the model, making it more robust in erasing concepts that are not explicitly described in the prompt.

To further support our claims, we conducted additional experiments on NSFW content erasure by fine-tuning the cross-attention modules. The results are presented in Table 8.

Table 8: Evaluation on the nudity erasure setting, where $-x$ and $-u$ denote fine-tuning the cross-attention and non-cross-attention modules, respectively.

| | NER-0.3↓ | NER-0.5↓ | NER-0.7↓ | NER-0.8↓ |
|---|---|---|---|---|
| SD | 16.69 | 10.91 | 5.46 | 2.02 |
| ESD-x | 10.25 | 5.83 | 2.17 | 0.68 |
| ESD-u | 5.32 | 2.36 | 0.74 | 0.23 |
| Ours-u | 3.64 | 1.70 | 0.40 | 0.06 |

It can be seen that the erasure performance by fine-tuning the non-cross-attention modules is significantly better than fine-tuning the cross-attention modules only, observed by the lower NER scores across all thresholds. This phenomenon is also observed in both the ESD and our method. Our method outperforms ESD in all settings by a large margin, demonstrating the effectiveness of our method in erasing NSFW content.

## B.3 Discussion on Metrics to Measure the Erasure Performance

One of the main challenges in developing erasure methods is the lack of a proper metric to measure erasure performance. Specifically, performance is evaluated by how well the model forgets the target concept while retaining other concepts. This raises a critical question: how can we validate whether a concept is present in a generated image? Although this may seem like a simple task, it is quite challenging due to the vast number of concepts that generative models can produce. It is infeasible to have a classification model capable of detecting all possible concepts.

While the FID score is a commonly used metric to assess the generative quality of models, it may not be sufficient for evaluating erasure performance. To the best of our knowledge, the CLIP alignment score is the most suitable existing metric for measuring concept inclusion. However, it is not without limitations. For example, CLIP's training set does not include NSFW content, making it less reliable for detecting such concepts. We believe that a more comprehensive evaluation metric is still lacking and that developing one would be a valuable direction for future research.

## B.4 Further Analysis on Searching for Adversarial Concepts

To further understand how our method searches for adversarial concepts, we provide intermediate results of the search process in Figure 6. The experiment is conducted on the Imagenette dataset with the same settings as in Section 5. Specifically, we simultaneously erase five concepts: "Garbage truck", "Cassette player", "Parachute", "Church", and "French horn".

In Figure 6, we show the images generated from the most sensitive concepts $c_a$ found by our method in the odd rows, as well as the corresponding to-be-erased concepts in the even rows. It is worth noting that all images are generated from the same initial noise input $z_T$, resulting in a similar background while still containing the target concepts, as shown in the first column of the even rows.

**The Removal Effect Through Fine-Tuning Steps.** As shown in the even rows, we observe that the model gradually removes the to-be-erased objects from the generated images as the fine-tuning steps increase. Interestingly, these to-be-erased concepts **tend to collapse into the same concept**, even though they started from different concepts. For example, the "Garbage truck" and "Cassette player" in the $2^{\text{nd}}$ and $4^{\text{th}}$ rows eventually transform into a background-like image in the last column. This can be explained by the fact that in the objective function 4, the erasing loss uses the same null concept $c_n$ for all to-be-erased concepts, which encourages the model to remove them simultaneously, eventually leading to the collapse of these concepts into the same form. This phenomenon can be an interesting direction for future research to investigate the relationship between different concepts in the erasing process, and the benefits of using different null concepts for different to-be-erased concepts.

**The Adversarial Concepts Adapt Through Fine-Tuning Steps.** On the other hand, the images generated from the most sensitive concepts $c_a$ in the odd rows show how they adapt to the erasing process. Interestingly, while the adversarial concepts $c_a$ can vary in each fine-tuning step—for example, the adversarial concept for "Garbage truck" in the first row changes from "truck", to "title", to "morning", and converges to "great" in the last column—the generated images $G(\theta'_t, z_T, c_a)$ change smoothly through the increasing fine-tuning steps $t$. This can be explained by the continuous update of the model $\theta'_t$ in each fine-tuning step, making $G(\theta'_t, z_T, \text{"truck"})$ and $G(\theta'_{t+1}, z_T, \text{"title"})$ are smoothly connected. This smooth transition of the generated images from the adversarial concepts $c_a$ demonstrates an advantage of our method, which allows for finding visual adversarial concepts rather than sticking to specific keywords.

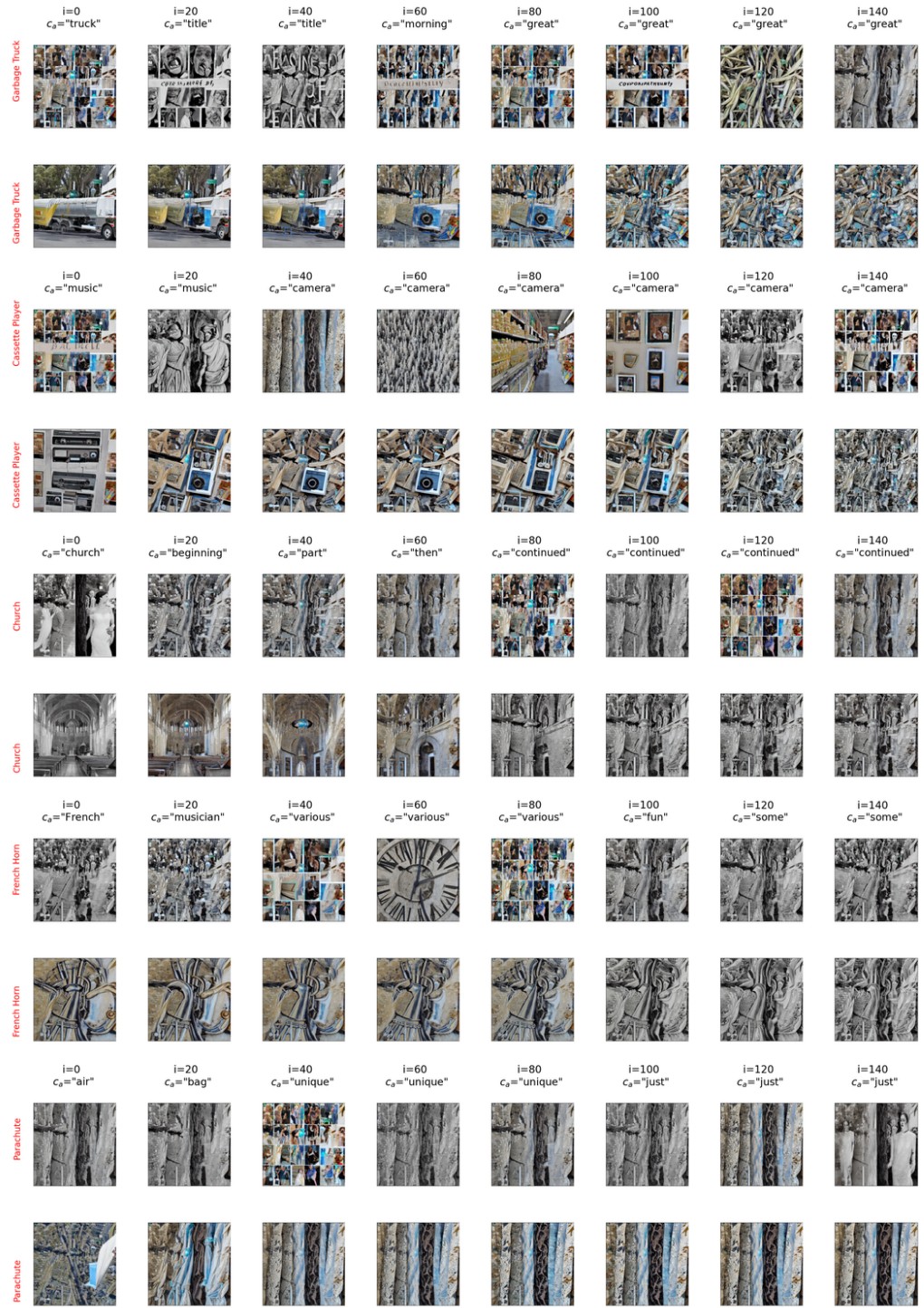

Figure 6: Intermediate results of the search process. Row-1,3,5,7,9: images generated from the most sensitive concepts $c_a$ found by our method. Row-2,4,6,8,10: images generated from the corresponding to-be-erased concepts. Each column represents different fine-tuning steps in increasing order.

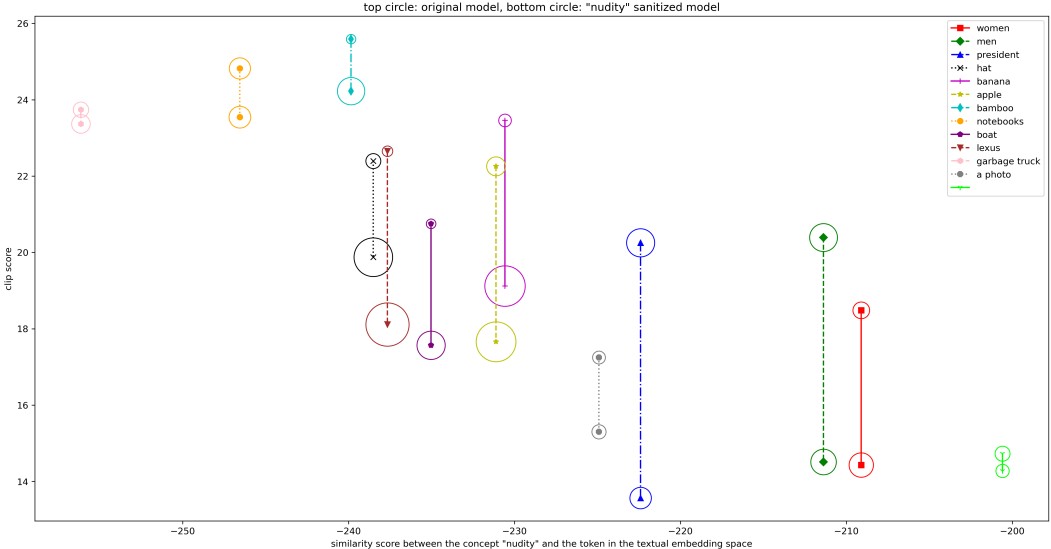

Figure 7: The figure shows the correlation between **the drop of the CLIP scores** (measured between generated images and their prompts) between the base/original model, and the sanitized model (i.e., removing the target concept "nudity") and **the similarity score** between the target concept "nudity" and other concepts in **the textual embedding space**. The radius of the circle indicates the variance of the CLIP scores measured in 200 samples, i.e., the larger circle indicates the larger variance of the CLIP scores.

## B.5 Difficulties in Searching for Adversarial Concepts

In this section, we provide empirical examples to show that finding the most sensitive concept $c_a$ is not always straightforward when using heuristic methods, which further emphasizes the advantage of our method.

**Can we use the similarity in the textual embedding space to find the most sensitive concept?**
Large pretrained multimodal models like CLIP have been widely used for zero-shot learning because their textual embedding space is highly correlated with the visual space. Intuitively, one might think that the similarity between the target concept $c_e$ and other concepts in the textual embedding space can help identify the most sensitive concept $c_a$. For example, the closer a concept $c_i$ is to the target concept $c_e$ in the textual embedding space, the more likely it is to be the most sensitive concept $c_a$. However, we demonstrate that this heuristic method is not always effective.

We conducted a similar analysis as in Section 3.2, including the similarity score between the target concept $c_e$ and other concepts in the textual embedding space to rank the concepts. Figure 7 shows the correlation between the drop in the CLIP scores between the base/original model and the sanitized model (i.e., after removing the target concept "nudity") and the similarity score between the target concept "nudity" and other concepts in the textual embedding space.

It can be seen that the above intuition does not always hold, as the similarity score does not correlate with the drop in the CLIP scores. For example, except for the concept "naked", the null concept is the most similar to "nudity" in the textual embedding space, but it experiences the lowest drop in CLIP scores. On the other hand, two concepts, "a photo" and "president", are close in the textual embedding space but are affected differently during the erasing process. This demonstrates that similarity in the textual embedding space is not an appropriate metric for identifying the most sensitive concept in this context.

## C  Qualitative Results

In addition to the quantitative results presented in Section 5, we provide qualitative results in this section to further demonstrate the effectiveness of our method compared to the baselines. Due to our

internal policy on publishing sensitive content, we are only able to show examples from two settings: erasing object-related concepts and erasing artistic concepts.

**Erasing Concepts Related to Physical Objects**  Figures 9, 10, and 11 show the results of erasing object-related concepts using ESD, UCE, and our method, respectively. Figure 8 shows the generated images from the original SD model. Each column represents different random seeds, and each row displays the generated images from either the to-be-erased objects or the to-be-preserved objects.

From Figure 8, we can see that the original SD model can generate all objects effectively. When erasing objects using ESD (Figure 9), the model maintains the quality of the preserved objects, but it also generates objects that should have been erased, such as the "Church" in the second row. This aligns with the quantitative results in Table 1, where ESD achieves the lowest erasing performance.

When using UCE (Figure 10), the model effectively erases the objects as shown in rows 1-5, but the quality of the preserved objects is significantly degraded, such as "tench" and "English springer" in the 8th and 9th rows. This is consistent with the quantitative results in Table 1, where UCE achieves the highest erasing performance but the lowest preservation performance.

In contrast, our method (Figure 11) effectively erases the objects while maintaining the quality of the preserved objects.

**Erasing Artistic Concepts**  Figures 12, 13, and 14 show the results of erasing artistic style concepts using our method compared to the baselines. Each column represents the erasure of a specific artist, except the first column, which represents the generated images from the original SD model. Each row displays the generated images from the same prompt but with different artists. The ideal erasure should result in changes in the diagonal pictures (marked by a red box) compared to the first column, while the off-diagonal pictures should remain the same. The results demonstrate that our method effectively erases the artistic style concepts while maintaining the quality of the remaining concepts.

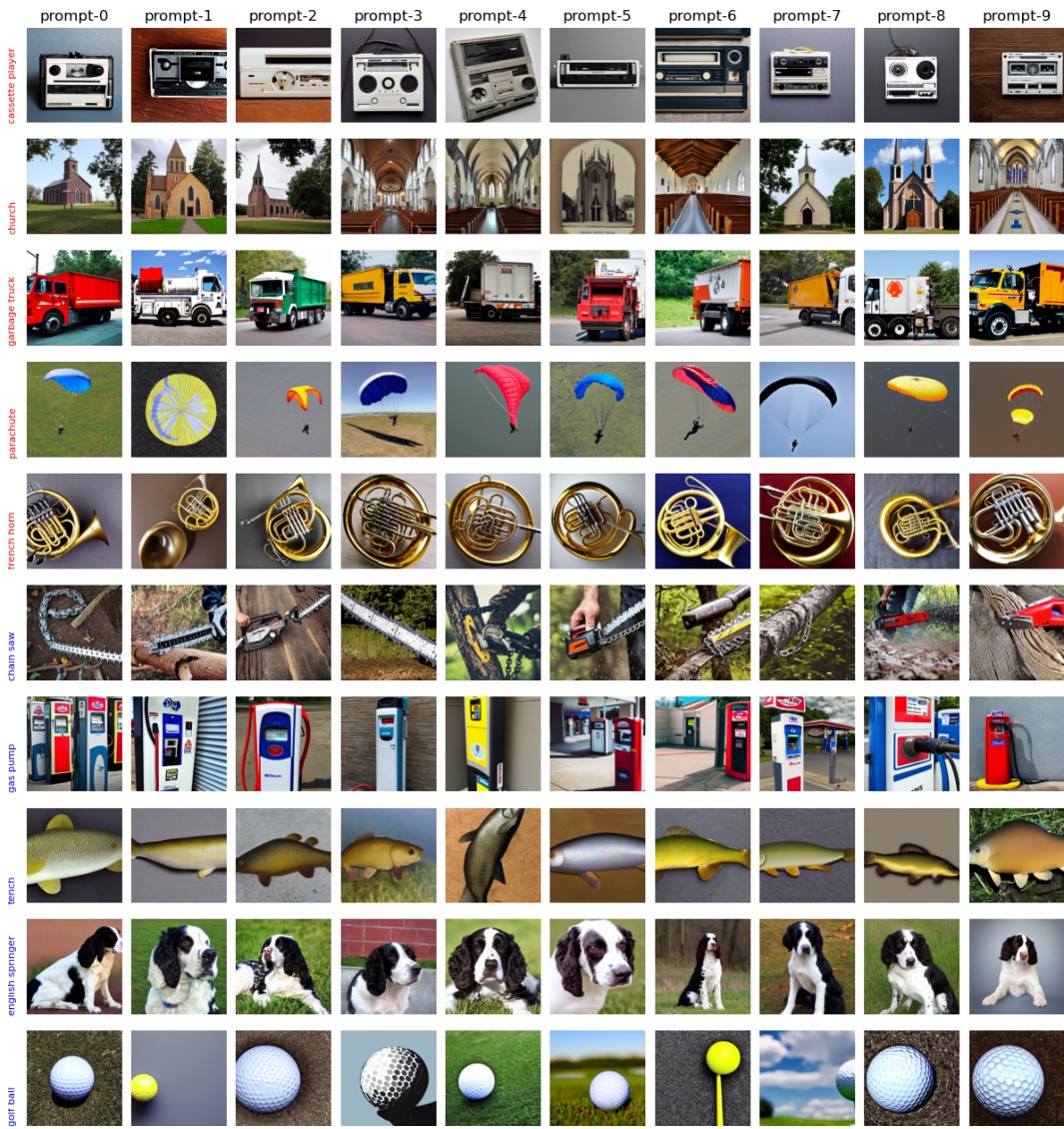

Figure 8: Generated images from the original model. Five first rows are to-be-erased objects (marked by red text) and the rest are to-be-preserved objects. Each column represents different random seeds.

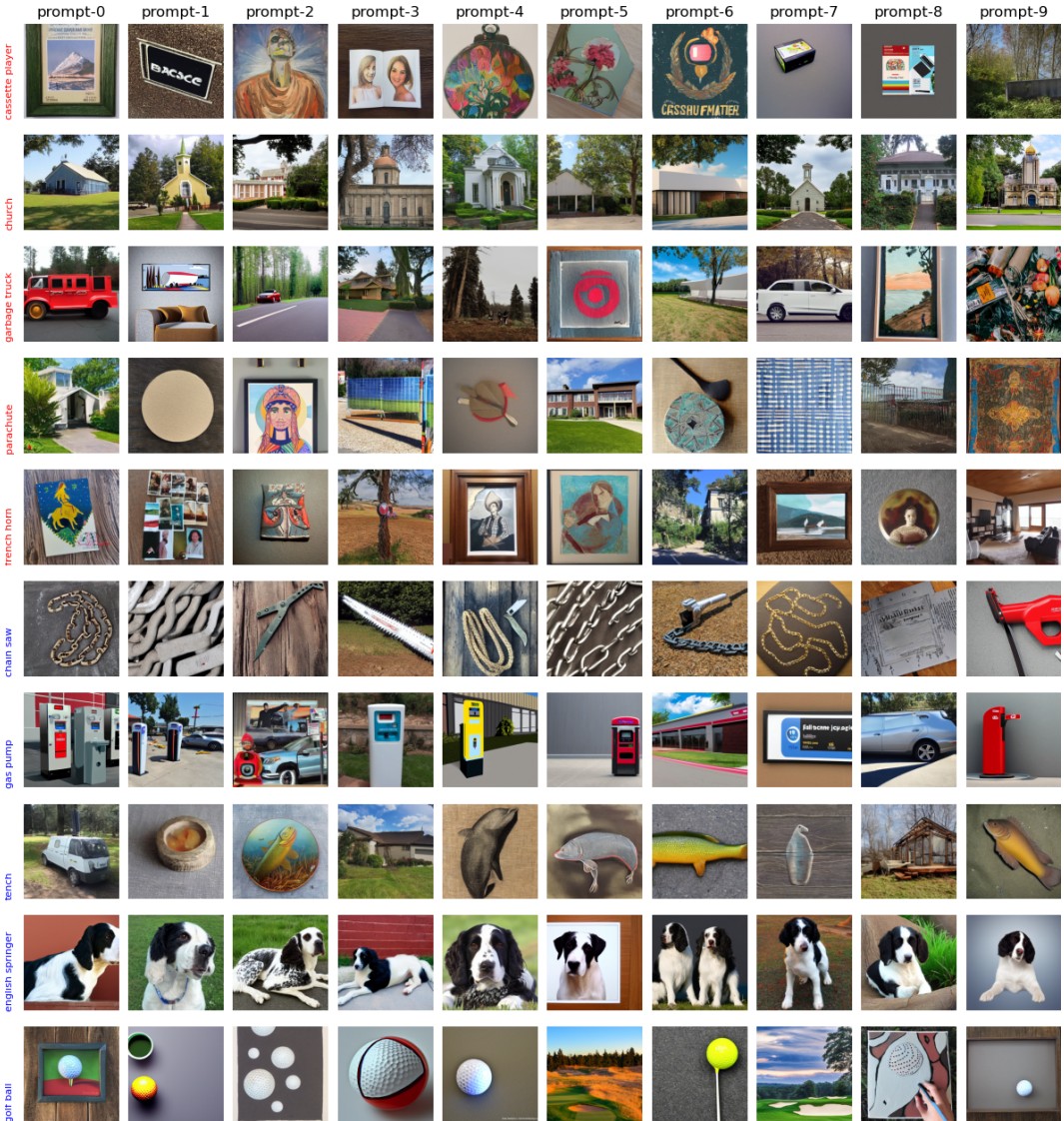

Figure 9: Erasing objects using ESD. Five first rows are to-be-erased objects (marked by red text) and the rest are to-be-preserved objects. Each column represents different random seeds.

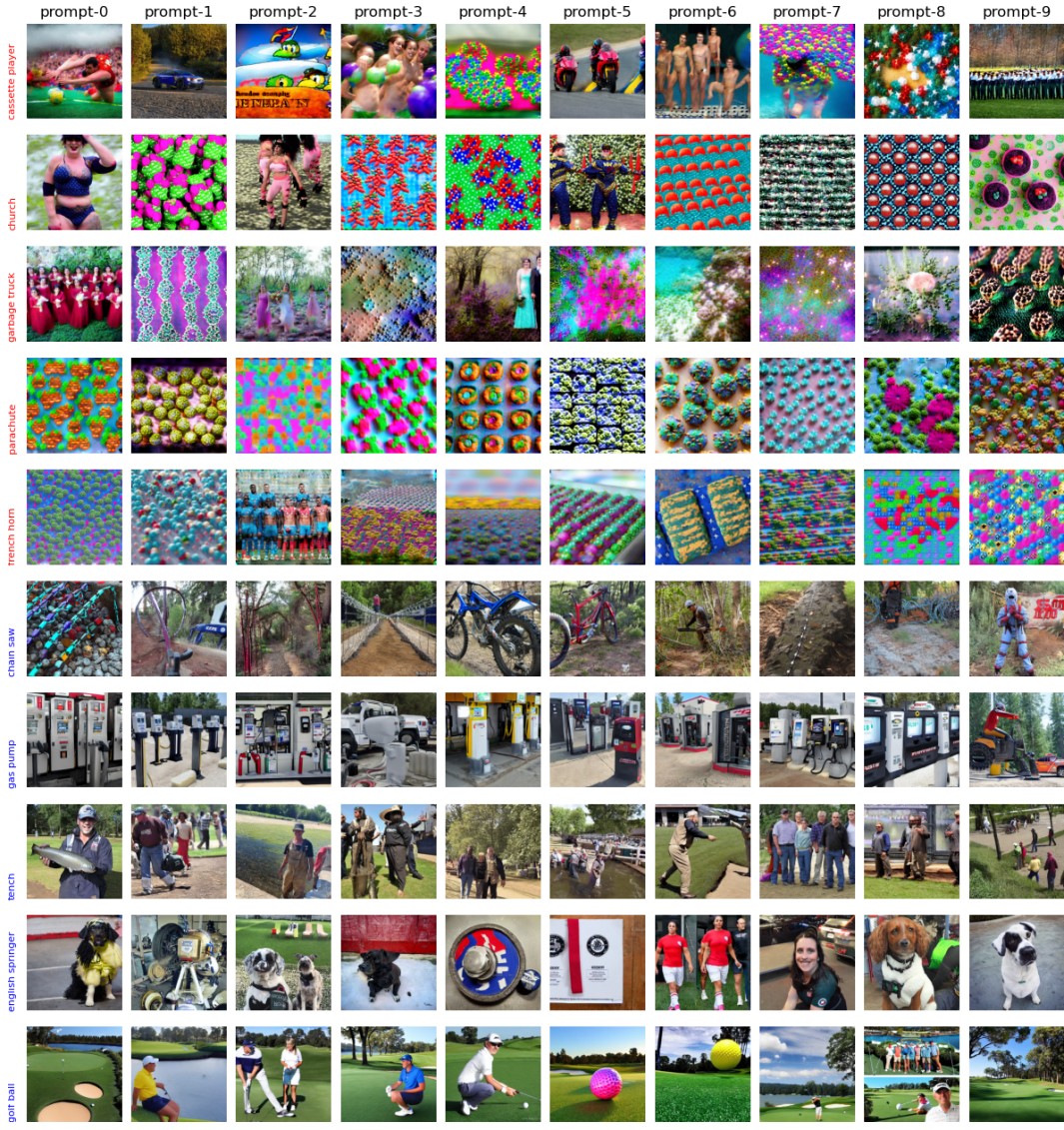

Figure 10: Erasing objects using UCE. Five first rows are to-be-erased objects (marked by red text) and the rest are to-be-preserved objects. Each column represents different random seeds.

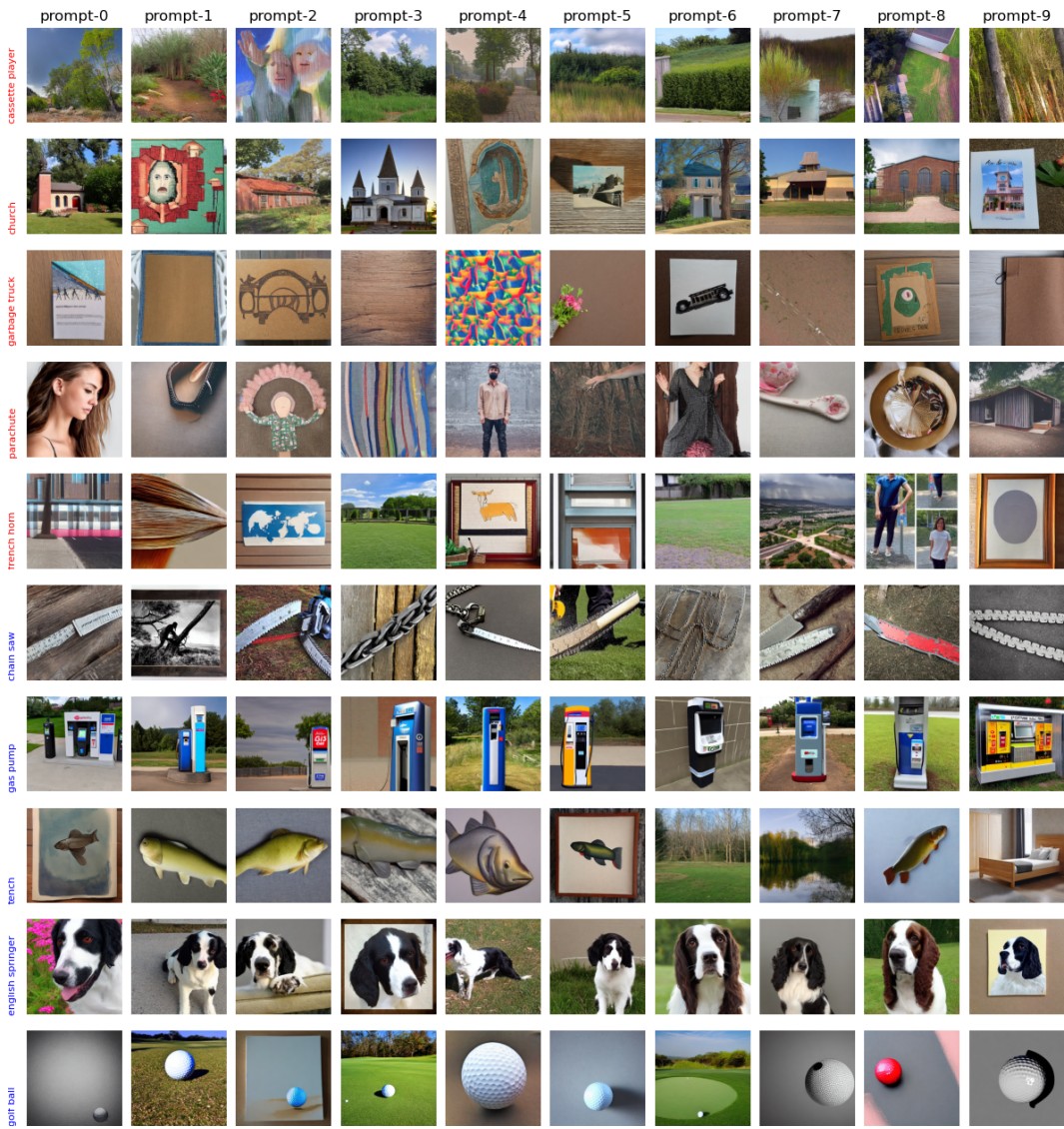

Figure 11: Erasing objects using our method. Five first rows are to-be-erased objects (marked by red text) and the rest are to-be-preserved objects. Each column represents different random seeds.

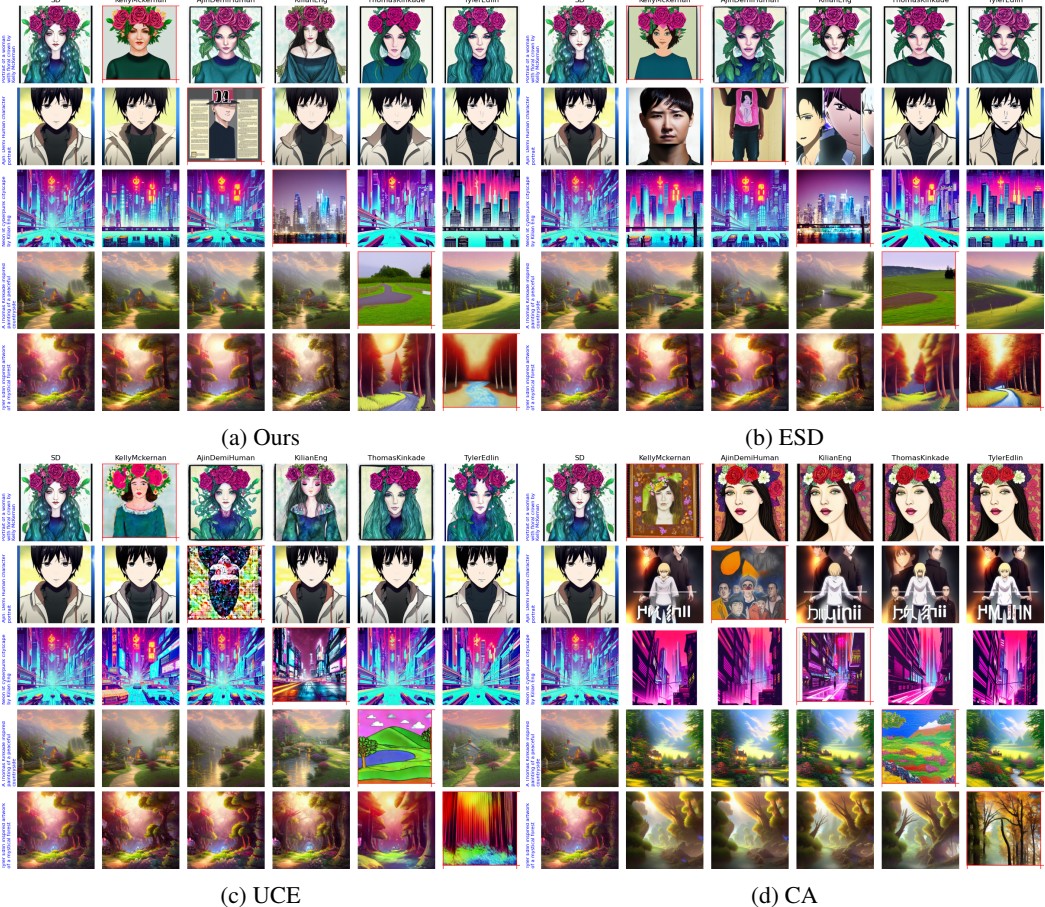

Figure 12: Erasing artistic style concepts. Each column represents the erasure of a specific artist, except the first column which represents the generated images from the original SD model. Each row represents the generated images from the same prompt but with different artists. The ideal erasure should result in the change in the diagonal pictures (marked by a red box) compared to the first column, while the off-diagonal pictures should remain the same. row-1: Portrait of a woman with floral crown by Kelly McKernan; row-2: Ajin: Demi Human character portrait; row-3: Neon-lit cyberpunk cityscape by Kilian Eng; row-4: A Thomas Kinkade-inspired painting of a peaceful countryside; row-5: Tyler Edlin-inspired artwork of a mystical forest;

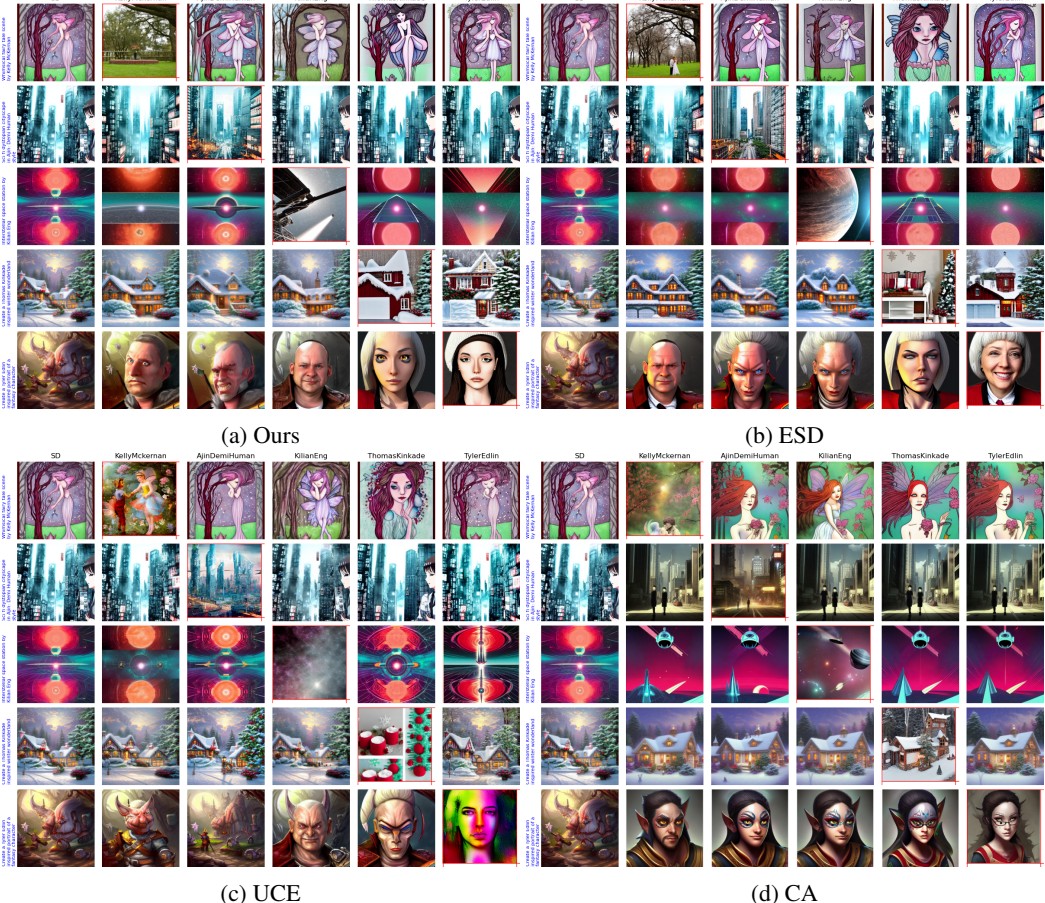

Figure 13: Erasing artistic style concepts (continue). Each column represents the erasure of a specific artist, except the first column which represents the generated images from the original SD model. Each row represents the generated images from the same prompt but with different artists. The ideal erasure should result in a change in the diagonal pictures (marked by a red box) compared to the first column, while the off-diagonal pictures should remain the same. row-1: Whimsical fairy tale scene by Kelly McKernan; row-2: Sci-fi dystopian cityscape in Ajin: Demi Human style; row-3: Interstellar space station by Kilian Eng; row-4: Create a Thomas Kinkade-inspired winter wonderland; row-5: Create a Tyler Edlin-inspired portrait of a fantasy character;

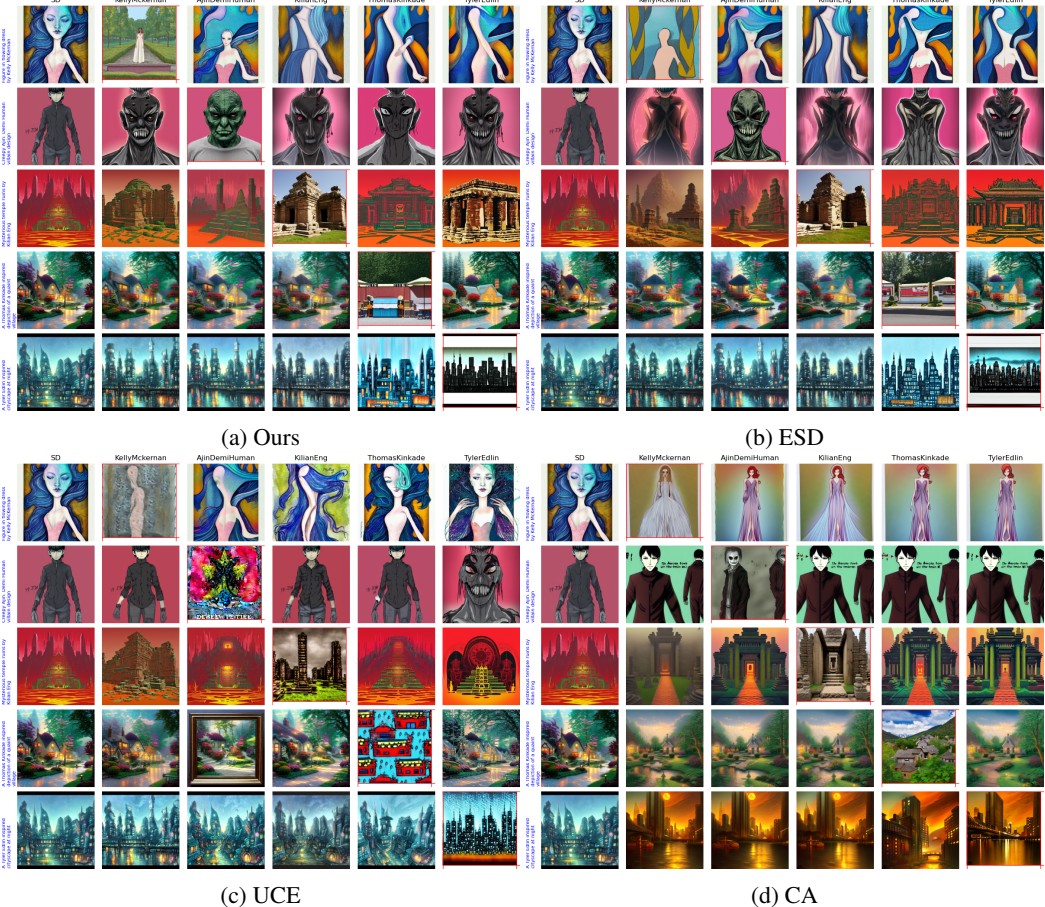

(a) Ours

(b) ESD

(c) UCE

(d) CA

Figure 14: Erasing artistic style concepts (continue). Each column represents the erasure of a specific artist, except the first column which represents the generated images from the original SD model. Each row represents the generated images from the same prompt but with different artists. The ideal erasure should result in a change in the diagonal pictures (marked by a red box) compared to the first column, while the off-diagonal pictures should remain the same. row-1: Figure in flowing dress by Kelly McKernan; row-2: Creepy Ajin: Demi Human villain design; row-3: Mysterious temple ruins by Kilian Eng; row-4: A Thomas Kinkade-inspired depiction of a quaint village; row-5: A Tyler Edlin-inspired cityscape at night;

