# OpenReview forum: "Erasing Undesirable Concepts in Diffusion Models with Adversarial Preservation"
_NeurIPS.cc/2024/Conference — NeurIPS 2024 poster_

### Official Review · Reviewer_fKJG · 2024-07-04

**Soundness:** 4
**Presentation:** 4
**Contribution:** 3
**Rating:** 7
**Confidence:** 5

**Summary:**

This paper investigates erasing undesirable concepts from stable diffusion. The paper builds up on a common finding on related works, that is removing even one concept can significantly rescue model’s ability to generate other concepts. Existing methods typically select a neutral concept, such as "a photo" or an empty string, as an anchor to preserve while erasing the target concept, expecting that maintaining the neutral concept should help retain other concepts as well (UCE, TIME). However, even regularization attempts using these neutral concept still affects the desirable concepts. This paper first find most sensitive concept for erasure, and then using a min-max (adversarial) adds another term to the loss function to remove only undesirable concepts.

**Strengths:**

Paper is well written and has a great flow. Authors clearly state the problem, current solutions and their drawback. They build their approach on these drawback and support their proposed method using different experiments.

**Weaknesses:**

- wrong citation for UCE in line 23
- Lines 29-30 are written as a strong conjecture without support. "no specific part of the model’s weights is solely responsible for a single concept". I have seen counter examples in the literature. See [1] for example
- While proposed method is showing improvement over related works, I doubt the practical usage and hence raise some questions.




[1] Basu, Samyadeep, et al. "On Mechanistic Knowledge Localization in Text-to-Image Generative Models." Forty-first International Conference on Machine Learning. 2024.

**Questions:**

- Which CLIP model is used?
- Why using a CLIP alignment score is a reliable measure for concept inclusion? For example a CLIP model that is trained on non-NSFW or non-GORE data cannot detect nudity/violence. What would your proposed method rely as a capability measure then?
- Line 152: the results of erasing the nudity concept, provided in the Appendix... I could not see this results. Best is also to have reference to specific plot/section in the appendix whenever you refer to appendix.
- Line 255: finetuning the non-cross-attention modules. I belive that including more details here is crucial. Why do you finetune only non-cross attention modules?
- Line 275: Did you see that FID is sufficient as a quality measure of generated samples when removing concepts? For example, removing nudity might distort anatomy and is not reflected in FID?
- Figure 4: We know that for nudity there is a huge difference between Feet and female breast for example. How would your proposed method only remove the later while keeping the non-sensitive parts untouched?
- What is the level of granularity in the proposed method? e.g. can we use this method to remove Mercedes logo from cars?
- How does your proposed method differ to "Forget-Me-Not" paper [1]? Any specific reason this is not covered in comparisons?
- I am curious to see what you think of task vectors [2] as a potential direction to remove undesirable concepts?

> code
- Code: train_adversarial_gumbel.py - lines 461-490: I could not see the implementation of equation (4) in these lines. Should not it be "loss += criteria(z_n_wo_prompt_pred.to(devices[0]), z_0_org_pred.to(devices[0]))" for L1 and "loss = -negative_guidance * criteria(z_r_wo_prompt_pred.to(devices[0]), z_r_org_pred.to(devices[0]))" for L2?
- When using L2 distance for similarity between vocab and the erasure wor, why do you use K-means? Why not using the top-n from the similarity result? did you look into result differences between these two?
- In the case of Stable Diffusion 3 (and SDXL), we have to have pooled and non-pooled captions. How would you calculate emb_r for the pooled embedding needed to  condition on time?

[1] Zhang, Gong, et al. "Forget-me-not: Learning to forget in text-to-image diffusion models." Proceedings of the IEEE/CVF Conference on Computer Vision and Pattern Recognition. 2024.
[2] Ilharco, Gabriel, et al. "Editing models with task arithmetic." arXiv preprint arXiv:2212.04089 (2022).

**Limitations:**

already addressed as questions for further discussions

---

> ### Author Rebuttal · Authors · 2024-08-07
>
> We thank the reviewer for the positive feedback and insightful suggestions. We would like to address the remaining concerns as follows.
> Due to the space limitation, some responses are provided in the global rebuttal and the author comment section.
>
> **Q: Why using a CLIP alignment score is a reliable measure for concept inclusion?**
>
> In Section 3.2, we investigated which concepts are affected most by the erasure of a target concept.
> The concept space that we investigate is not only a specific set of related concepts as shown in Figure 1 but also a broader set of concepts of the CLIP token vocabulary which includes 49,408 tokens as shown in Figure 2.
> In this general and broader concept space, it is nearly impossible to have a pre-trained classification model that can detect all concepts.
> Therefore, to the best of our knowledge, the CLIP alignment score might be the most effective way to measure the concept inclusion,
> even though it is not perfect and might not be able to provide a reliable measure for some concepts as suggested by the reviewer.
> We will add this discussion to the revised version.
>
> **Q: Results of erasing nudity concept"**
>
> We apologize for the confusion. We would like to provide the results of erasing the "nudity" concept while preserving the "person" concept in the attached document.
> As shown in the figure, preserving related concepts like "person" helps to retain the model's capability on other concepts much better than preserving a neutral concept.
>
> **Q: Why do you finetune only non-cross attention modules to erase NSFW concepts?**
>
> Firstly, we would like to recall the cross-attention mechanism, i.e., $\sigma(\frac{(QK^T)}{\sqrt{d}})V$, where $Q$, $K$, and $V$ are the query, key, and value matrices, respectively.
> In text-to-image diffusion models like SD, the key and value are derived from the textual embedding of the prompt, while the query comes from the previous denoising step.
> The cross-attention mechanism allows the model to focus on the relevant parts of the prompt to generate the image.
>
> Therefore, when unlearning a concept, most of the time, the erasure process is done by loosening the attention between the query and the key that corresponds to the concept to be erased, i.e., by fine-tuning the cross-attention modules.
> This approach works well for object-related concepts or artistic styles, where the target concept can be explicitly described with limited textual descriptions.
>
> However, as investigated in the ESD paper Section 4.1, concepts like 'nudity' or NSFW content can be described in various ways, many of which do not contain explicit keywords like 'nudity.'
> This makes it inefficient to rely solely on keywords to indicate the concept to be erased.
> It is worth noting that the standard SD model has 12 transformer blocks, each of which contains one cross-attention module but also several non-cross-attention modules such as self-attention and feed-forward modules, not to mention other components like residual blocks.
> Therefore, fine-tuning the non-cross-attention modules will have a more global effect on the model, making it more robust in erasing concepts that are not explicitly described in the prompt.
> We will add more explanations to the revised version.
>
> **Q: Discussion on FID metric?**
>
> We thank the reviewer for raising a very important question.
>
> Firstly, we utilized FID measured on common concepts like the COCO-30K dataset to evaluate the generation quality of the sanitized models because it is a widely used metric in the field of generative models.
> However, while this metric is useful in general generation settings, we acknowledge that it might not be sufficient for evaluating unlearning concepts.
>
> As observed in our paper, the impact of erasure methods is not equally distributed across all concepts; some concepts might be more affected than others.
> While this imbalance might be mitigated with a large enough evaluation size that covers all possible concepts, making FID sufficient, this is not the case in practice, which commonly uses the COCO-30K dataset.
>
> Therefore, we believe that a more comprehensive evaluation metric that can capture the impact of erasing concepts on specific concepts is needed.
> For example, we could assign a weight to each evaluated concept/prompt based on its relevance score to the target concepts, then compute a weighted FID or CLIP score.
>
> **Q: How would your proposed method only remove the sensitive parts while keeping the non-sensitive parts untouched in the NSFW setting?**
>
> While our method has demonstrated an interesting result in its ability to remove more sensitive body parts, we do not have a specific explanation for this phenomenon. However, intuitively speaking, if we consider two abstract concepts, "nudity" and "person," and two core concepts, "breast" and "feet," we can see that the "nudity" concept might be highly correlated with the "breast" concept, while the "person" concept might be more correlated with the "feet" concept rather than "breast."
>
> Therefore, if we specifically preserve the "person" concept while erasing the "nudity" concept, the model might be able to generate images that include "feet" but exclude "breast." Our method, by selecting the most affected concepts, might naturally choose a concept that is highly correlated with non-sensitive body parts to be preserved.
>
> **Q: What is the level of granularity in the proposed method?**
>
> Our method does not have a specific mechanism to control the level of granularity or focus on a specific concept to be erased. Instead, our main goal is to prioritize the preservation aspect of the erasure process. However, if we were to address the problem of granularity, we would focus on enhancing the expressiveness of the textual description.
>
> For example, using visual embeddings to describe the Mercedes logo concept could be a potential direction to improve the expressiveness and granularity of the method.

---

> > ### Comment · Reviewer_fKJG · 2024-08-12
> > **answer to Authors rebuttal**
> >
> > Thank you for taking the time to address my concerns. Unfortunately some of my questions are still not addressed and I really hope that we can discuss over those to conclude this rebuttal. Please see my responses below
> >
> >     Q: reliability of CLIP alignment
> > I believe that limitations of using CLIP score should be clearly mentioned in your draft and you need to focus more on non-NSFW removal because of limitation mentioned before.
> >
> >     Q: Results of erasing nudity concept
> > I could not find these results. When adding any new figures/plots provide details how to find them and make the life of reader easier. This is also my strong recommendation when writing your draft. You need to add details which figure in Appendix you are referring to.
> >
> >     Q: Why do you finetune only non-cross attention modules to erase NSFW concepts?
> > Do you have any results/references to ESD that supports/compares cross-attention finetuning vs. all attention finetuning?
> >
> >
> >     Q: Discussion on FID metric?
> > I believe that running FID is a good automatic evaluation but not sufficient. You should always look at generated samples and include them. (examples in the appendix are good enough)
> >
> >     Q: Granularity/only remove the sensitive parts while keeping the non-sensitive parts
> > My main concern, that is not reflected in limitations is the practical usage of the proposed method. When asking these question, I was hoping to get some non-intuitive responses, such as evidences to support the strength of the proposed method or at least discussions on the future directions.
> >
> >     Q: "Forget-Me-Not" paper
> > I think that the comparison with this work should be included in the main manuscript. I was surprised not to see this in the comparisons. Please provide any results of you have tried single concept with this method.
> >
> >     Q: Coding question: Implementations of Equation 4 and Equation 5
> > I spent sometime to map your equations in the paper with the provided anonymous code when writing the original review, and hence asked specific details questions about any potential bug in your implementation. Please provide more details that might remove my/readers confusions.
> >
> >     Q: SDXL/SD3 implementation
> > I still do not understand how to get the pooled embedding. in both of these methods we need pooled and non-pooled features. I can see that non-pooled can be optimized using gumbel-softmax but your response still does not cover the pooled caption.

---

> > > ### Author Response · Authors · 2024-08-12
> > > **Further responses (1/n)**
> > >
> > > We thank the reviewer for actively engaging in the discussion and providing valuable feedback. In the following, we would like to address the reviewer's further comments:
> > >
> > > **Q: I believe that limitations of using CLIP score should be clearly mentioned in your draft and you need to focus more on non-NSFW removal because of limitations mentioned before.**
> > >
> > > We appreciate the reviewer's feedback. We will discuss the limitations of using the CLIP alignment score in the revised version.
> > >
> > > **Q: Results of erasing nudity concept**
> > >
> > > We appreciate the reviewer's recommendation and deeply apologize for forgetting to provide the details of the figure in the main paper and the appendix.
> > > We provided the results of erasing the "nudity" concept while preserving the "person" concept in the attached document in the global rebuttal.
> > > More specifically, we compare the impact of erasing the same "nudity" to other concepts with different preserving strategies, including preserving a fixed concept such as " ", "person", and the most affected concepts found by our method.
> > > We will add a reference to the specific figure in the revised version.
> > >
> > > **Q: Do you have any results/references to ESD that supports/compares cross-attention finetuning vs. all attention finetuning?**
> > >
> > > In addition to the explanation provided in the previous response, we would like to provide additional experiments with the ESD method with different fine-tuning strategies as below.
> > > More specifically, we compare the erasure performance of the ESD method by fine-tuning the cross-attention modules only (ESD-x) and fine-tuning non-cross-attention modules only (ESD-u).
> > >
> > > |          | NER-0.3↓ | NER-0.5↓ | NER-0.7↓ | NER-0.8↓ |
> > > |----------|----------|----------|----------|----------|
> > > | SD       | 16.69    | 10.91    | 5.46     | 2.02     |
> > > | ESD-x    | 10.25    | 5.83     | 2.17     | 0.68     |
> > > | ESD-u    | 5.32     | 2.36     | 0.74     | 0.23     |
> > > | Ours-u   | 3.64     | 1.70     | 0.40     | 0.06     |
> > >
> > > It can be seen that the erasure performance by fine-tuning the non-cross-attention modules is significantly better than fine-tuning the cross-attention modules only,
> > > observed by the lower NER scores across all thresholds.
> > >
> > > The detailed results at threshold 0.5 are shown in the table below, with the number of exposed parts and the number of images with any exposed parts.
> > >
> > > |                        | SD  | ESD-x | ESD-u |
> > > |------------------------|-----|-------|-------|
> > > | Feet                   | 92  | 61    | 24    |
> > > | Belly                  | 212 | 81    | 21    |
> > > | Armpits                | 261 | 123   | 63    |
> > > | Buttocks               | 53  | 26    | 3     |
> > > | Male Breast            | 75  | 27    | 14    |
> > > | Male Genitalia         | 23  | 13    | 9     |
> > > | Female Genitalia       | 28  | 7     | 1     |
> > > | Female Breast          | 331 | 124   | 38    |
> > > | Total #exposed part    | 1075| 462   | 173   |
> > > | Total #img-with-any-expose | 513 | 274 | 111   |
> > > | NER                    | 10.91| 5.83 | 2.36  |
> > >
> > > **Q: I believe that running FID is a good automatic evaluation but not sufficient. You should always look at generated samples and include them. (examples in the appendix are good enough)**
> > >
> > > We thank the reviewer for the suggestion. Because of the page limit, we could not include the generated samples in the main paper.
> > > We will try to include them in the revised version or provide a clear reference to the appendix.

---

> > > > ### Author Response · Authors · 2024-08-12
> > > > **Further responses (2/n)**
> > > >
> > > > **Q: Granularity/only remove the sensitive parts while keeping the non-sensitive parts**
> > > >
> > > > We thank the reviewer for the insightful questions. We will add further discussions on the practical usage of the proposed method. As highlighted in the main paper, our primary goal is to focus on the preservation aspect of the erasure process. To the best of our knowledge, our method is the first to examine the impact of erasing a target concept on other concepts and to propose an adaptive strategy for preserving the most affected concepts to maintain the model's performance on other concepts.
> > > >
> > > > ***Regarding the less intuitive responses***, we would like to provide the similarity scores between different concepts and body parts in the nudity erasure setting as below.
> > > >
> > > > |                        | Nudity | A photo | Person | Body  |
> > > > |------------------------|--------|---------|--------|-------|
> > > > | Feet                   | 0.612  | 0.547   | 0.566  | 0.643 |
> > > > | Belly                  | 0.601  | 0.514   | 0.517  | 0.748 |
> > > > | Armpits                | 0.614  | 0.477   | 0.475  | 0.643 |
> > > > | Buttocks               | 0.649  | 0.501   | 0.494  | 0.639 |
> > > > | Male Breast            | 0.616  | 0.499   | 0.472  | 0.504 |
> > > > | Male Genitalia         | 0.618  | 0.511   | 0.537  | 0.517 |
> > > > | Female Genitalia       | 0.662  | 0.536   | 0.558  | 0.555 |
> > > > | Female Breast          | 0.656  | 0.517   | 0.491  | 0.574 |
> > > >
> > > > It can be seen that the "nudity" concept is highly correlated with the "Female Breast" concept, suggesting that when removing the "nudity" concept, the "Female Breast" concept is more likely to be affected than other body parts. On the other hand, the "Person" or "Body" concept is more strongly correlated with the "Feet" concept than with the "Female Breast" concept, indicating that preserving the "Person" concept might help maintain the model's performance on "Feet" rather than on "Female Breast."
> > > >
> > > > Furthermore, the gap between the "Feet" and "Female Breast" concepts with respect to "Person" or "Body" is larger than the gap with more generic concepts like "A photo." This suggests that preserving generic concepts might not have the same impact as preserving the most affected concepts.
> > > >
> > > > Our method naturally selects the most affected concepts to be preserved, which often includes concepts highly correlated with non-sensitive body parts. This explains the observed phenomenon in the experiment.
> > > >
> > > > ***Regarding potential directions to improve the granularity of the method***, we would like to provide two potential directions:
> > > >
> > > > - Enhancing the expressiveness of the textual description: By providing more detailed textual descriptions, the model can distinguish between closely related concepts, such as separating the Mercedes logo from the car itself.
> > > > For example, in Equation 4 and Equation 5, instead of using the textual embedding $c_e = \tau(\text{"Mercedes logo"})$, we could use the Textual Inversion method to learn a visual embedding $c_e = TI(X_{c_e})$ from the representative images of the Mercedes logo concept $X_{c_e}$.  This approach provides a richer expressiveness to specify the target concept to be erased.
> > > >
> > > > - Searching for multiple affected concepts: In the current approach, we search for the most affected concept to be preserved for each target concept. This approach is simple and fast but of course, has limited representative capability.
> > > > Intuitively, each target concept connects to many other concepts, and choosing several affected concepts can provide a more comprehensive perspective on the impact of erasing the target concept.
> > > > It can be done by applying SVGD (Stein Variational Gradient Descent) to search for multiple affected concepts, which encourages the method to explore a diverse set of concepts with high relevance to the target concept.
> > > >
> > > > **Q: "Forget-Me-Not" paper**
> > > >
> > > > We will include a comparison with the Forget-Me-Not method in the revised version. For this comparison, we followed the instructions provided in their repository to set up experiments for erasing single concepts from the Imagenette dataset. Unfortunately, we were unable to effectively erase the target concepts using this method.
> > > >
> > > > Specifically, we followed these steps:
> > > >
> > > > - Provided representative images of the target concepts (8 images per concept) to train the inversion models.
> > > > - Executed the inversion process using the ti_config.yaml configuration file.
> > > > - Conducted the erasure process with the attn.yaml configuration file.
> > > > - Generated and stored images with the erased concepts in the evaluation_folder/exps_attn directory.
> > > >
> > > > We have uploaded the code for this experiment to the anonymous repository. You can find the config files, run file, and generated images in the Forget-Me-Not/evaluation_folder/exps_attn folder. The generated images still contain the target concepts, except for the "parachute" concept, indicating that the Forget-Me-Not method was not effective for erasing the target concepts in our setting. Please refer to the code for additional details.

---

> > > > > ### Comment · Reviewer_fKJG · 2024-08-13
> > > > > **response to authors**
> > > > >
> > > > > I thank the authors for working hard, adding additional experiments and addressing all my concerns. I would like to increase my score by 2, now to 7.

---

> > > > > > ### Author Response · Authors · 2024-08-13
> > > > > > **Thank you the reviewer**
> > > > > >
> > > > > > We greatly appreciate the reviewer's supportive feedback and insightful comments, which have not only significantly improved our paper but also pointed out aspects we had not considered. We will incorporate the suggested discussions into the revised version. Once again, thank you for your time and effort in reviewing our work.

---

> ### Author Response · Authors · 2024-08-07
> **Further responses**
>
> **Q: Which CLIP model is used?**
>
> We used the OpenAI CLIP model `openai/clip-vit-large-patch14' to compute the alignment score. We will add this information to the revised version.
>
> **Q: How does your proposed method differ to the "Forget-Me-Not" paper? Any specific reason this is not covered in comparisons?**
>
> Our method significantly differs from the Forget-Me-Not (FMN) method. From our point of view, FMN falls into the category of approaches like TIME, UCE, and MACE,
> which focuses on confusing the alignment between the prompt and the visual features in the cross-attention mechanism.
> Specifically, FMN introduces an attention resteering method that attempts to alter the attention maps related to the target concept (i.e., by minimizing the L2 norm of the attention maps).
> Our method, on the other hand, stands out by focusing on identifying which concepts are most affected by the erasure of a target concept and then preserving these concepts to maintain the model's capability on other concepts.
> We will cite and discuss the FMN in the revised version.
>
> Regarding additional experiments with the FMN method, we attempted to erase the same set of concepts from the Imagenette dataset and evaluated the erasing and preservation performance of the FMN method within our setting.
> However, despite our best efforts given the time constraints, FMN did not effectively erase the target concepts.
> Notably, their open-source code provides hyper-parameter settings for single-concept erasure but not for multiple concepts.
> We also noticed an open issue on their GitHub repository questioning this same problem.
>
> This issue can easily be verified by running their code with a modified configuration file, `attn.yaml`,
> to erase multiple concepts, given some representative images of the target concepts.
>
> **Q: Coding question: Implementations of Equation 4 and Equation 5**
>
> Equation 4 describes our naive approach, which utilizes Projected Gradient Descent (PGD) to search for the adversarial concepts in the continuous space. We have uploaded the code for this approach to the anonymous repository. Please refer to the code for more details.
>
> Regarding the implementation of Equation 5, our method involves bilevel optimization. In this process, the inner maximization step maximizes the L2 loss (i.e., minimizes -L2 loss) to select the most affected concepts to be preserved. The outer minimization step minimizes the combined L1 + L2 loss to erase the target concepts while preserving the most affected concepts.
>
> For the L1 loss, we inherited this implementation from the ESD paper and did not attempt to modify it, including the aspect of negative guidance.
>
> **Q: Coding question: Why using K-means?**
>
> In the implementation, we investigate the use of K-means to control the tradeoff between the computational cost and the size of the vocabulary when searching for adversarial concepts. More specifically, we first compute the similarity between the target concept and the entire vocabulary, then select the top-K most similar concepts. We then use K-means to choose the K most representative concepts from these top-K most similar concepts. This approach allows us to cover a wide range of concepts while keeping the computational cost low.
>
> **Q: Coding question: How to calculate the text embedding for adversarial concepts in Stable Diffusion version 3**
>
> To the best of our understanding, as described in the `encode_prompt` function in the Stable Diffusion v3 pipeline (lines 310-311),
> as similar to the same function in the SDXL pipeline (line 397), we still can obtain the standard prompt embedding as the output of a text encoder which is independent of the time step. Therefore, we can still use our current approach to calculate the `emb_r` for the adversarial concepts.

---

> ### Author Response · Authors · 2024-08-12
> **Looking forward to your responses!**
>
> As the rebuttal period is coming to an end, we hope to have clarified the novelty of our contribution and addressed the concerns of the reviewer. We truly appreciate the reviewer's time on our paper and we are looking forward to your responses and/or any other potential questions. Your feedback would be very helpful for us to identify any ambiguities in the paper for further improvement.

---

### Official Review · Reviewer_Zhhp · 2024-07-04

**Soundness:** 3
**Presentation:** 3
**Contribution:** 3
**Rating:** 7
**Confidence:** 4

**Summary:**

The present paper addresses the challenge of erasing content from text-to-image diffusion models with a focus on reducing the degenerative impact on other concepts. To this end, the authors propose a novel approach that focuses on identifying and preserving adversarial concepts—those which are most affected by changes in model parameters during the erasure. To provide evidence, the authors conduct empirical investigations and various experiments using the open-source Stable Diffusion model, demonstrating their method’s potential to reliably erase target concepts while offering minimal impact on non-target concepts.

**Strengths:**

- The paper is well-structured, providing clear and comprehensive explanations of the proposed method and its theoretical foundations.
- The novelty of focusing on adversarial concepts to balance erasure and preservation is a significant contribution to the field.
The experiments are well selected. They first showcase the reliability of the proposed method on a generic task, followed by two relevant use cases, namely erasing unethical and artistic concepts.
- While not presented in the main paper, the authors address limitations of their proposed method in the appendix.

**Weaknesses:**

- While the paper showcases the benefits of the proposed method, the observations and conclusions drawn from the empirical experiments do not fully align with what is described by the authors, particularly regarding the minimal impact on other concepts. For example, in lines 231-232, the authors state that the proposed method achieves much higher ESR scores than the two baselines ESD and CA; however, this claim is not entirely accurate, especially in the case of CA. Similarly, the improvements over the baselines shown in the third experiment are not as significant as described, particularly when considering the standard deviations presented in Table 3. The paper would benefit from a more detailed discussion and analysis of these results.

**Questions:**

- As far as I understand, the set of concepts used to search for adversarial concepts is derived from the list of words in the initial prompt (lines 76-79). However, it appears to be a common set used across multiple examples. Can you provide more details and clarify how this set of concepts is selected? Have you experimented with different sets of initial concepts?

- Further, you mentioned in lines 252-255 that, in the case of mitigating unethical content, it is necessary to fine-tune the non-cross-attention modules. Can you elaborate on why this is the case?

Minor comments:
- In line 238, CA seems to be the best baseline, not ESD.

**Limitations:**

The authors addressed limitations in the appendix as stated in the checklist.

---

> ### Author Rebuttal · Authors · 2024-08-07
>
> We thank the reviewer for the positive feedback and insightful suggestions. We would like to address the remaining concerns as follows.
>
> **Q: Better discussion**
>
> We thank the reviewer for pointing this out. We will revise the comparison with CA in the revised version, i.e., our method is slightly better than CA in terms of ESR scores but significantly better in terms of PSR scores.
>
> **Q: How to choose the concept space**
>
>
> We did not design specific but utilized a common set of concepts as the search space $\mathcal{R}$ for searching adversarial concepts across all experiments.
> To ensure the generality of the search space so that it can be applied to various tasks such as object-related concepts, NSFW content, and artistic styles,
> we used the Oxford 3000 most common words in English as the search space.
>
> It is worth noting that as described in Section B.1, our method employ the Gumbel-Softmax trick to discretely search in the concept space $\mathcal{R}$,
> this approach requires feeding the model with the embeddings of the entire search space $\mathcal{R}$ to compute the alignment score,
> which is computationally expensive when the search space is large.
> To mitigate this, we use a subset of the K most similar concepts to reduce the computational cost.
>
> To better address the reviewer's concern, we conducted additional experiments with the search space as the CLIP token vocabulary, which includes 49,408 tokens.
> It is worth noting that the CLIP token vocabulary is more comprehensive but presents challenges due to the large number of nonsensical tokens (e.g., )
> Therefore, we need to filter out these nonsensical tokens to ensure the quality of the search space.
> The results from object-related concepts are shown in the table below.
>
> | Vocab  | ESR-1 ↑ | ESR-5 ↑ | PSR-1 ↑ | PSR-5 ↑ |
> |--------|---------|---------|---------|---------|
> | Oxford | 98.72   | 95.60   | 63.80   | 82.96   |
> | CLIP   | 97.88   | 94.80   | 69.24   | 87.20   |
>
> The results show that the erasing performance is slightly lower when using the CLIP token vocabulary as the search space,
> but the preservation performance is much better with a gap of 5.4\% in PSR-1 and 4.2\% in PSR-5.
> This indicates that our method would benefit from a more comprehensive search space.
> We will add this experiment to the revised version.
>
> **Q: it is necessary to fine-tune the non-cross-attention modules in erasing NSFW concept**
>
> Firstly, we would like to recall the cross-attention mechanism, i.e., $\sigma(\frac{(QK^T)}{\sqrt{d}})V$, where $Q$, $K$, and $V$ are the query, key, and value matrices, respectively.
> In text-to-image diffusion models like SD, the key and value are derived from the textual embedding of the prompt, while the query comes from the previous denoising step.
> The cross-attention mechanism allows the model to focus on the relevant parts of the prompt to generate the image.
>
> Therefore, when unlearning a concept, most of the time, the erasure process is done by loosening the attention between the query and the key that corresponds to the concept to be erased, i.e., by fine-tuning the cross-attention modules.
> This approach works well for object-related concepts or artistic styles, where the target concept can be explicitly described with limited textual descriptions.
>
> However, as investigated in the ESD paper Section 4.1, concepts like 'nudity' or NSFW content can be described in various ways, many of which do not contain explicit keywords like 'nudity.'
> This makes it inefficient to rely solely on keywords to indicate the concept to be erased.
> It is worth noting that the standard SD model has 12 transformer blocks, each of which contains one cross-attention module but also several non-cross-attention modules such as self-attention and feed-forward modules, not to mention other components like residual blocks.
> Therefore, fine-tuning the non-cross-attention modules will have a more global effect on the model, making it more robust in erasing concepts that are not explicitly described in the prompt.
> We will add more explanations to the revised version.

---

> ### Author Response · Authors · 2024-08-12
> **Looking forward to your responses!**
>
> As the rebuttal period is coming to an end, we hope to have clarified the novelty of our contribution and addressed the concerns of the reviewer. We truly appreciate the reviewer's time on our paper and we are looking forward to your responses and/or any other potential questions. Your feedback would be very helpful for us to identify any ambiguities in the paper for further improvement.

---

> > ### Comment · Reviewer_Zhhp · 2024-08-12
> >
> > Thank you for the rebuttal and for including the additional experiments, which have clarified some aspects of the search space design. I also appreciate the additional experiments shown in the global rebuttal.
> >
> > Regarding the claim to “fine-tune the non-cross-attention modules,” I agree with Reviewer fKJG that presenting results to support this claim would significantly strengthen the argument.

---

> > > ### Author Response · Authors · 2024-08-12
> > > **Further response**
> > >
> > > We thank the reviewer for actively engaging in the discussion and providing valuable feedback.
> > >
> > > Regarding the discussion on "fine-tuning the non-cross-attention modules": In addition to the explanation provided in the previous response, we would like to provide additional experiments with the ESD method with different fine-tuning strategies as below.
> > > More specifically, we compare the erasure performance of the ESD method by fine-tuning the cross-attention modules only (ESD-x) and fine-tuning non-cross-attention modules only (ESD-u).
> > >
> > > |          | NER-0.3↓ | NER-0.5↓ | NER-0.7↓ | NER-0.8↓ |
> > > |----------|----------|----------|----------|----------|
> > > | SD       | 16.69    | 10.91    | 5.46     | 2.02     |
> > > | ESD-x    | 10.25    | 5.83     | 2.17     | 0.68     |
> > > | ESD-u    | 5.32     | 2.36     | 0.74     | 0.23     |
> > > | Ours-u   | 3.64     | 1.70     | 0.40     | 0.06     |
> > >
> > > It can be seen that the erasure performance by fine-tuning the non-cross-attention modules is significantly better than fine-tuning the cross-attention modules only,
> > > observed by the lower NER scores across all thresholds.
> > >
> > > The detailed results at threshold 0.5 are shown in the table below, with the number of exposed parts and the number of images with any exposed parts.
> > >
> > > |                        | SD  | ESD-x | ESD-u |
> > > |------------------------|-----|-------|-------|
> > > | Feet                   | 92  | 61    | 24    |
> > > | Belly                  | 212 | 81    | 21    |
> > > | Armpits                | 261 | 123   | 63    |
> > > | Buttocks               | 53  | 26    | 3     |
> > > | Male Breast            | 75  | 27    | 14    |
> > > | Male Genitalia         | 23  | 13    | 9     |
> > > | Female Genitalia       | 28  | 7     | 1     |
> > > | Female Breast          | 331 | 124   | 38    |
> > > | Total #exposed part    | 1075| 462   | 173   |
> > > | Total #img-with-any-expose | 513 | 274 | 111   |
> > > | NER                    | 10.91| 5.83 | 2.36  |

---

> > > > ### Comment · Reviewer_Zhhp · 2024-08-14
> > > > **Score update**
> > > >
> > > > Thank you again for the effort. The provided clarifications will improve the soundness of the paper and its claims. I updated my score accordingly.

---

> > > > > ### Author Response · Authors · 2024-08-14
> > > > > **Thank you the reviewer**
> > > > >
> > > > > We greatly appreciate the reviewer for raising the score. We will incorporate the suggested discussions into the revised version. Once again, thank you for your time and effort in reviewing our work.

---

### Official Review · Reviewer_r1Xs · 2024-07-13

**Soundness:** 2
**Presentation:** 1
**Contribution:** 2
**Rating:** 3
**Confidence:** 4

**Summary:**

This paper focuses on the problem that existing concept erasing methods struggle to address the trade-off between the generation capability of erased concepts and remaining concepts. To address this problem, this paper proposes a method that erases the target concept while minimizing the impact of other concepts. Specifically, this paper finds that related concepts are sensitive during the erasing process of the target concept. For example, when erasing 'nudity', some related concepts, such as 'woman' and 'people' will be significantly impacted. Motivated by this observation, this paper first utilizes an optimization method to automatically find the adversarial concepts related to the target concept. Following this, this paper iteratively applies the preservation constraint on these adversarial concepts during the erasing process.

**Strengths:**

- This paper provides an empirical observation that removing different target concepts leads to varying impacts on other concepts, which is interesting and helpful for future research.
- In the experiment of erasing object-related concepts, the proposed method demonstrates the effectiveness in maintaining the generation capability of remaining concepts.

**Weaknesses:**

- The motivation of this paper needs further discussed. My core issue is: Do all related concepts need to be preserved? For example, if we want to erase 'airplane', do 'aircraft' and 'warplane' need to be preserved? Motivated by this issue, I argue that there should be a boundary between the preserved and erasing concepts, while this paper ignores this boundary. More extremely, some works [1,2,3] argue that we should erase concepts related to the target concept.
- Lack of comparison with the latest methods, such as MACE [4].
- In the experiment of erasing object-related concepts, it is suggested to add an experiment that demonstrates the generalization of the model. For example, following MACE [4], this paper can evaluate the erasing capability on the synonyms of the erasing concept.
- In the experiment of erasing NSFW content, this paper lacks an evaluation of the generation capability on the common concepts. Following [2,3], this paper can evaluate the FID of the model on COCO dataset.

**Questions:**

Please check weaknesses.

---

> ### Author Rebuttal · Authors · 2024-07-31
>
> We thank the reviewer for the insightful comments and suggestions. We would like to address the remaining concerns as follows.
> Due to the space limitation, some responses are provided in the global rebuttal.
>
> **Q: Do all related concepts need to be preserved? Boundary between preserved and erasing concepts**
>
> We agree with the reviewer that when erasing a target concept like 'airplane,' the synonyms that resemble it visually, such as 'aircraft,' should also be erased.
> However, while our method does not explicitly set a boundary when searching for the adversarial concepts, the optimization process in our method naturally selects the most affected concepts but not the `similar' concepts to the target concept to be preserved.
> We support our response with both theoretical and empirical evidence.
>
> Theoretical perspective:
>
> Our optimization framework, described in Equations 4 and 5, involves a bilevel optimization process.
> The outer level (w.r.t. $\theta'$) minimizes the erasing loss L1 and the preservation loss L2 simultaneously,
> while the inner level (w.r.t. $c_a$) maximizes L2 to select the most affected concepts to be preserved.
>
> Initially, $\theta' = \theta$, therefore after minimizing L1, the most affected concepts will be exactly the target concept $c_e$.
> As optimization progresses, the model $\theta'$ diverges from the original model $\theta$, driving the output $\epsilon_{\theta'}(c_e)$ away from $\epsilon_{\theta}(c_e)$ and closer to $\epsilon_{\theta}(c_n)$.
> This process makes the target concept $c_e$ one of the affected concepts or a candidate for the inner maximization w.r.t. $c_a$.
> However, selecting $c_a = c_e$ would directly conflict with the erasing loss L1 that aims to drive $c_e \rightarrow c_n$.
> Consequently, the inner maximization process steers towards selecting the most affected concepts but not the target concept $c_e$ or its synonyms.
>
> Empirical perspective:
>
> As mentioned in Appendix B.4 (lines 835-844) and shown in Figure 12, the intermediate results of the adversarial concept selection process indicate that the model initially selects concepts similar to the target. For instance, Figure 12 shows that 'truck,' 'music,' 'church,' 'French,' and 'bag' are selected as adversarial concepts for 'Garbage truck,' 'Cassette player,' 'Church,' 'French horn,' and 'Parachute,' respectively. Over time, the model shifts towards less similar but highly affected concepts.
>
> Moreover, our response to another question demonstrates that our method effectively erases both the target concept and its synonyms, as shown by the results of erasing the synonyms.
>
> In summary, both theoretical and empirical evidence support that our method naturally prioritizes preserving the most affected concepts, which are not the synonyms of the target concept but those significantly impacted by its erasure.
>
> **Q: Compare to MACE**
>
> Following the reviewer's suggestion, we conducted an additional experiment using MACE with their official implementation on object-related concepts, as detailed in section 5.1 of the main paper.
> Specifically, we conducted four distinct tasks, each involving the erasure of five concepts from the Imagenette dataset.
> In addition, we evaluated the preservation performance with common concepts from the COCO-30K dataset with FID and CLIP scores.
> Due to time constraints, we were only able to complete one task, which involved erasing five concepts: Cassette Player, Church, Garbage Truck, Parachute, and French Horn.
>
> The results are presented in Table 1 in the attached document.
> Regarding erasing performance, MACE slightly outperformed our method by 0.7\% in ESR-1 and 0.5\% in ESR-5 on average.
> However, in terms of preservation performance, our method significantly outperformed MACE, with a gap of 8\% in PSR-1 and 7\% in PSR-5.
> Additionally, our method achieved superior preservation performance in image generation with common concepts, evidenced by the lowest FID score of 16.3 and the highest CLIP score of 26.1.
>
> These results indicate that while MACE shows marginally better erasing performance, our method excels significantly in preservation performance, even when compared to the latest techniques.
>
> **Q: Evaluating FID on COCO dataset in erasing NSFW setting**
>
> We already measured the FID on the COCO dataset and provided the results in Table 2 in the main paper.

---

> ### Author Response · Authors · 2024-08-12
> **Looking forward to your responses!**
>
> As the rebuttal period is coming to an end, we hope to have clarified the novelty of our contribution and addressed the concerns of the reviewer. We truly appreciate the reviewer's time on our paper and we are looking forward to your responses and/or any other potential questions. Your feedback would be very helpful for us to identify any ambiguities in the paper for further improvement.

---

### Official Review · Reviewer_Jwe1 · 2024-07-13

**Soundness:** 2
**Presentation:** 3
**Contribution:** 2
**Rating:** 4
**Confidence:** 4

**Summary:**

This paper studies the memory-forgetting tradeoff for concept removal. The authors systematically summarize the tradeoff problem, and propose the idea of adversarial concepts to solve it. Specifically, this approach automatically detects the most sensitive concepts that will be affected by unlearning, and enforces the model to maintain the performance of sensitive concepts by adding the maintaining loss.

**Strengths:**

1. This paper is well-written

2. The summary of the performance drop problem for unlearning is systematic.

**Weaknesses:**

1. Results are mainly on SD1.4. Additional results on other stable diffusion models should be included. For example, on larger diffusion models like SD XL, to further validate the effectiveness of the proposed method.
2. In the experiment section, comparisons with existing state-of-the-art methods, such as SPM [1], are lacking.
3. The ESR-k and PSR-k metrics used by the authors are reasonable and widely adopted by previous methods. However, additional metrics, such as FID and CLIP Score, should also be included to demonstrate the model's performance in the object unlearning scenario. Similarly, evaluating the FID metric on COCO-30K is also necessary.
4. When visualizing results in the appendix, the authors should show images before and after unlearning to provide a more intuitive sense of performance. Specifically, changes in unrelated concepts should be minimal. The current images only indicate that the concept has not been forgotten, rather than showing that the generated images under the prompt have not changed.
5. Efficiency is also an important evaluation metric in machine unlearning. The authors should adequately compare the efficiency of their method with previous methods.
6. When comparing methods, the authors should contrast their preservation methods with previous methods, such as the modules used in ConAbl [2] and SPM [1], to highlight performance differences.

[1] One-dimensional Adapter to Rule Them All: Concepts, Diffusion Models, and Erasing Applications
[2] Ablating Concepts in Text-to-Image Diffusion Models

**Questions:**

How to select the parameter $\lambda$? It seems that different concepts may need different coefficients.

**Limitations:**

Yes

---

> ### Author Rebuttal · Authors · 2024-08-07
>
> We thank the reviewer for the insightful comments and suggestions. We would like to address the remaining concerns as follows.
> Due to the space limitation, some responses are provided in the author's comment section.
>
> **Q: Additional experiments with SOTA methods such as ConAbl and SPM, and evaluation with FID and CLIP scores on COCO-30K dataset.**
>
> We followed the reviewer's suggestion and conducted additional experiments with MACE, a SOTA method recently accepted at CVPR 2024.
> It is worth noting that we already compared with the suggested baseline ConAbl denoted as CA in our paper.
>
> Specifically, we performed the object-related setting which includes four tasks, each involving the erasure of five concepts from the Imagenette dataset.
> In addition to ESR-1, ESR-5, PSR-1, and PSR-5 metrics, we generated images with common concepts from the COCO-30K dataset and assessed FID and CLIP scores.
> Due to time constraints, we completed one task, erasing five concepts: Cassette Player, Church, Garbage Truck, Parachute, and French Horn.
>
> The results are presented in the table below.
>
> | Method |  | ESR-1 ↑ |  | ESR-5 ↑ |  | PSR-1 ↑ |  | PSR-5 ↑ |  | FID ↓ |  | CLIP ↑ |  | FT Time |
> |--------|--|---------|--|---------|--|---------|--|---------|--|-------|--|--------|--|---------|
> | ESD    |  | 95.5 ± 0.8 |  | 88.9 ± 1.0 |  | 41.2 ± 12.9 |  | 56.1 ± 12.4 |  | 17.9  |  | 24.5   |  | 40 mins |
> | UCE    |  | 100 ± 0.0  |  | 100 ± 0.0  |  | 23.4 ± 3.6  |  | 49.5 ± 8.0  |  | 19.1  |  | 21.4   |  | 4 mins  |
> | CA     |  | 98.4 ± 0.3 |  | 96.8 ± 6.1 |  | 44.2 ± 9.7  |  | 66.5 ± 6.1  |  | 16.6  |  | 25.8   |  | 12 mins |
> | MACE   |  | 99.3 ± 0.3 |  | 97.6 ± 1.2 |  | 47.4 ± 12.0 |  | 72.8 ± 10.5 |  | 16.9  |  | 24.9   |  | 3 mins  |
> | Ours   |  | 98.6 ± 1.1 |  | 96.1 ± 2.7 |  | 55.2 ± 10.0 |  | 79.9 ± 2.8  |  | 16.3  |  | 26.1   |  | 65 mins |
>
> Regarding erasing performance, MACE slightly outperformed our method by 0.7% in ESR-1 and 0.5% in ESR-5 on average.
> However, in terms of preservation performance, our method significantly outperformed MACE, with an 8% gap in PSR-1 and a 7% gap in PSR-5.
> Additionally, our method achieved superior preservation performance in image generation with common concepts, evidenced by the lowest FID score of 16.3 and the highest CLIP score of 26.1.
>
> These results indicate that while MACE shows marginally better erasing performance, our method excels significantly in preservation performance, even when compared to the latest techniques.
>
> **Compared to SPM:**
>
> The SPM method, although effective in concept erasure, does not directly fine-tune the original model.
> Instead, it trains a separate adapter that, when attached to the original model, prevents the generation of the erased concept.
> Specifically, Section 3.1 of the SPM paper introduces a new diffusion process $\hat{\epsilon} = \epsilon(x_t, c, t \mid \theta, \mathcal{M}_{c_e})$
>
> , where $\mathcal{M}_{c_e}$ is the adapter model trained to erase the concept $c_e$.
> While these adapters can be shared and reused across different models, the original model $\theta$ remains unchanged, allowing malicious users to generate the erased concepts easily.
> In contrast, our method, along with the other baselines in our paper, directly erases the concept from the original model, making it more robust and preventing the generation of the erased concepts.
> For this reason, we did not include SPM in the comparison.
>
> **Q: Additional experiments with larger models**
>
> Due to resource constraints in the short rebuttal timeframe, we were only able to conduct additional experiments on SD v1.4. However, it is worth noting that SD v1.4 is still the most widely used model in the community, including recent work such as MACE.
> We will consider conducting experiments on larger models like SD XL in future work.
>
> **Q: Evaluating Efficiency**
>
> Following the reviewer's suggestion, we provided the fine-tuning time for the object-related concepts task in the table above.
> It is worth noting that we could only find efficiency evaluation in the SPM paper, which is specifically designed to highlight the efficiency of their lightweight adapter approach.
> Our method which was designed to prioritize erasing and preserving performance, may not be as efficient as SPM in terms of fine-tuning time.
> However, we politely argue that efficiency is not the main concern in our work or other baselines, as the fine-tuning process is relatively fast, typically completing in less than a few hours, which is acceptable in practice for infrequent concept erasure requests.
>
> **Q: Evaluating the impact of $\lambda$**
>
> To investigate the impact of different $\lambda$ values, we conducted additional experiments on the object-related concepts task with $\lambda = 0.1, 0.5, 5, 10$.
> The results are presented in the table below.
> There is a clear trade-off between erasing and preserving performance when changing the $\lambda$ value.
> A smaller $\lambda$ value results in better erasing performance but worse preservation performance, and vice versa.
> In our experiments, we did not attempt to tune $\lambda$ for each concept but just simply set it to 1 for all experiments as mentioned in line 774.
>
> | λ     | ESR-1 ↑ | ESR-5 ↑ | PSR-1 ↑ | PSR-5 ↑ |
> |------------|---------|---------|---------|---------|
> | 0.1    | 97.88   | 94.52   | 29.44   | 40.72   |
> | 0.5    | 98.28   | 94.32   | 56.04   | 73.92   |
> | 1      | 98.72   | 95.60   | 63.80   | 82.96   |
> | 5      | 96.96   | 91.68   | 74.84   | 91.52   |
> | 10     | 91.64   | 84.48   | 83.04   | 96.64   |

---

> ### Author Response · Authors · 2024-08-07
> **Further response**
>
> **Q: Better visualization**
>
> In Figures 9-11 in the Appendix, we already provided a comparison between the output generated by the same prompt using the original model and the sanitized models after erasing each specific artistic style.
>
> In each sub-figure, the first column shows the images generated by the original model, while the second to sixth columns display the images generated by the sanitized models. Each row corresponds to a prompt of one of these five artists.
>
> The ideal erasure should result in changes in the diagonal pictures (marked by a red box) compared to the first column, while the off-diagonal pictures should remain the same.
> We believe that these visualizations are clear and informative, providing a direct comparison between the original and sanitized models.

---

> ### Author Response · Authors · 2024-08-12
> **Looking forward to your responses!**
>
> As the rebuttal period is coming to an end, we hope to have clarified the novelty of our contribution and addressed the concerns of the reviewer. We truly appreciate the reviewer's time on our paper and we are looking forward to your responses and/or any other potential questions. Your feedback would be very helpful for us to identify any ambiguities in the paper for further improvement.

---

### Author Rebuttal · Authors · 2024-08-07

We thank all the reviewers for their insightful comments and suggestions. Below are our responses to some important questions raised by the reviewers. We kindly request the reviewers to consider raising the scores if our responses adequately address the remaining concerns.

**Q: Compare to MACE a SOTA method**

Following the reviewer's suggestion, we conducted an additional experiment using MACE (accepted to CVPR 2024) with their official implementation on object-related concepts, as detailed in section 5.1 of the main paper.
Specifically, we conducted four distinct tasks, each involving the erasure of five concepts from the Imagenette dataset.
In addition, we evaluated the preservation performance with common concepts from the COCO-30K dataset with FID and CLIP scores.
Due to time constraints, we were only able to complete one task, which involved erasing five concepts: Cassette Player, Church, Garbage Truck, Parachute, and French Horn.

The results are presented in Table 1 in the attached document and below

| Method | ESR-1 ↑       | ESR-5 ↑       | PSR-1 ↑       | PSR-5 ↑       | FID ↓ | CLIP ↑ |
|--------|---------------|---------------|---------------|---------------|-------|--------|
| ESD    | 95.5 ± 0.8    | 88.9 ± 1.0    | 41.2 ± 12.9   | 56.1 ± 12.4   | 17.9  | 24.5   |
| UCE    | 100 ± 0.0     | 100 ± 0.0     | 23.4 ± 3.6    | 49.5 ± 8.0    | 19.1  | 21.4   |
| CA     | 98.4 ± 0.3    | 96.8 ± 6.1    | 44.2 ± 9.7    | 66.5 ± 6.1    | 16.6  | 25.8   |
| MACE   | 99.3 ± 0.3    | 97.6 ± 1.2    | 47.4 ± 12.0   | 72.8 ± 10.5   | 16.9  | 24.9   |
| Ours   | 98.6 ± 1.1    | 96.1 ± 2.7    | 55.2 ± 10.0   | 79.9 ± 2.8    | 16.3  | 26.1   |

Regarding erasing performance, MACE slightly outperformed our method by 0.7\% in ESR-1 and 0.5\% in ESR-5 on average.
However, in terms of preservation performance, our method significantly outperformed MACE, with a gap of 8\% in PSR-1 and 7\% in PSR-5.
Additionally, our method achieved superior preservation performance in image generation with common concepts, evidenced by the lowest FID score of 16.3 and the highest CLIP score of 26.1.

These results indicate that while MACE shows marginally better erasing performance, our method excels significantly in preservation performance, even when compared to the latest techniques.

**Q: Evaluating performance on erasing synonyms**

We follow the suggestion to evaluate the erasing capability on the synonyms of object-related concepts, e.g., "Church".
More specifically, we first utilize a set of tools including ChatGPT, Dictionary/Thesaurus.com, and Google image search to find the best synonyms for each target concept.
To verify that these synonyms are indeed resembling the target concept, we then use the original model to generate images from the synonyms (e.g., "a photo of Chapel"),
and using the ResNet-50 model to classify the generated images.
We then only keep the synonyms that have the top-5 accuracy higher than 50\% to ensure that they are indeed generation-similar to the target concept.
To this end, for some concepts, we could not find any good synonyms such as "Golf ball" or "Chain saw" except for some minor variations.
We provide (top-1 and top-5) accuracy of the synonyms as below, as well as those numbers of target concepts, the higher the accuracy, the more similar the synonyms are to the target concept. Due to the space constraint, we are only able to provide the numbers of 6/10 concepts.

- **Church (84.4;100.0)**: chapel (80.0;100.0), cathedral (50.0;100.0), minster (87.5;100.0), basilica (32.5;100.0)
- **Garbage truck (83.2;99.2)**: trash truck (87.5;97.5), refuse truck (80.0;100.0), waste collection vehicle (97.5;100.0), sanitation truck (47.5;100.0)
- **Parachute (95.2;99.2)**: skydiving chute (93.9;100.0), paraglider (100.0;100.0)
- **Chain saw (76.4;89.0)**: chainsaw (92.0;96.0), power saw (26.0;58.0)
- **Tench (76.0;98.0)**: cyprinus tinca (60.0;95.0), cyprinus zeelt (52.5;100.0)
- **Golf ball (98.2;99.2)**: golfing ball (99.0;99.0)

Given the list of `valid' synonyms, we then generate images from the synonyms using the sanitized models obtained from four object-related settings (each set corresponds to erasing five Imagenette concepts simultaneously).

The results are shown in Table 2 in the attached document and as below:

| Method | $\text{ESR}_{s}$-1 ↑ | $\text{ESR}_{s}$-5 ↑ | $\text{PSR}_{s}$-1 ↑ | $\text{PSR}_{s}$-5 ↑ |
|--------|----------------------|----------------------|----------------------|----------------------|
| SD-org | 22.0 ± 11.6          | 2.4 ± 1.4            | 78.0 ± 11.6          | 97.6 ± 1.4           |
| SD-syn | 41.5 ± 8.2           | 7.5 ± 2.3            | 58.5 ± 8.2           | 92.5 ± 2.3           |
| UCE    | 99.8 ± 0.1           | 99.2 ± 0.5           | 19.5 ± 4.4           | 43.8 ± 0.6           |
| MACE   | 98.1 ± 0.9           | 84.7 ± 2.2           | 41.4 ± 10.3          | 73.3 ± 3.1           |
| Ours   | 85.2 ± 6.1           | 72.3 ± 7.1           | 46.6 ± 6.5           | 82.9 ± 4.5           |

Firstly, if comparing SD-org and SD-syn (the original/target concept and the synonyms),
we can see that the top-1 accuracy of SD-syn is significantly lower than that of SD-org, but the top-5 accuracy is lower `only' by 5\% on average.
This does not mean that the model could not generate meaningful images from the synonyms, but rather the images are not recognized as the target concept by the ResNet-50 model in top-1 prediction but still in top-5.

Secondly, MACE is the best method in terms of erasing synonyms, followed by our method.
However, in trade-off, our method is better in preserving performance, not only the original concepts as shown in the main paper but also the synonyms.

Thirdly, we should also consider the FID and CLIP scores to evaluate the preservation performance of the model, in which our method is much better than MACE.

---

### Decision · Program_Chairs · 2024-09-25

**Decision:**

Accept (poster)

**Comment:**

This paper studies methods to erase certain concepts from diffusion models, specifically focusing on NSFW and artistic concepts as a main use case. The challenge here is that if certain concepts are removed, then highly correlated ones could be impacted (e.g., removing 'nudity' may impact general generation of people). Compared to prior work, this paper focuses on an adversarial learning approach, looking at concepts that are most affected by parameter changes. The authors also investigate the trade-off between removing some concepts while actively trying to not adversely impact other concepts. This differentiates the present work from other baselines (e.g., Forget-Me-Not) because prior work focuses more on erasing concepts, rather than the trade-off between erasure and preservation.

This is a tough paper to evaluate. The reviewers all agree that this is a very important and timely problem to study, as diffusion models are becoming more wide-spread, while still not being very well understood in terms of undesirable concept generation. Reviewers have several complaints about the comparison to different methods (e.g., MACE), the inclusion various metrics (e.g., FID), and way the conclusions are drawn from the experimental results. For the last point, a core criticism is that the authors have not thoroughly discovered where the concepts originate from (e.g., different layers of the model vs. the latent embeddings). However, the authors have addressed many of these negative points in the appendix and in the rebuttal.

The reviewers that vote for acceptance are effectively championing the paper, where they have also had a very thorough back-and-forth with the authors during the rebuttal period. Namely, the main selling point is that the paper tries to address an important problem, and make a real world impact. The reviewers that vote for rejection raise valid points. However, I think that the importance and inherent complexity of the problem means that we cannot expect clear-cut evaluations, and hence, it is worth accepting a paper that will have impact by inspiring more nuanced follow-up work. The space of concept erasure in diffusion models is also not especially crowded at the moment, and therefore, the present work does present a lot of novel findings, research frameworks, and empirical insights compared to the related work.

Overall, I recommend acceptance.

For the authors, please add experiments comparing to forget-me-not, adding experiments for SDXL (if possible) and include much more details in the appendix and main paper to address the comments from the reviewers. Also, as a way to address the negative reviews, please consider rephrasing some of the findings in the abstract/intro to be more clear about the findings of the paper. For example it is too broad to say that "This approach ensures stable erasure with minimal impact on the other concepts."